# Are GATs Out of Balance?

**Nimrah Mustafa**[*]
nimrah.mustafa@cispa.de

**Aleksandar Bojchevski**[†]
a.bojchevski@uni-koeln.de

**Rebekka Burkholz**[*]
burkholz@cispa.de

[*]CISPA Helmholtz Center for Information Security, 66123 Saarbrücken, Germany
[†]University of Cologne, 50923 Köln, Germany

## Abstract

While the expressive power and computational capabilities of graph neural networks (GNNs) have been theoretically studied, their optimization and learning dynamics, in general, remain largely unexplored. Our study undertakes the Graph Attention Network (GAT), a popular GNN architecture in which a node's neighborhood aggregation is weighted by parameterized attention coefficients. We derive a conservation law of GAT gradient flow dynamics, which explains why a high portion of parameters in GATs with standard initialization struggle to change during training. This effect is amplified in deeper GATs, which perform significantly worse than their shallow counterparts. To alleviate this problem, we devise an initialization scheme that balances the GAT network. Our approach i) allows more effective propagation of gradients and in turn enables trainability of deeper networks, and ii) attains a considerable speedup in training and convergence time in comparison to the standard initialization. Our main theorem serves as a stepping stone to studying the learning dynamics of positive homogeneous models with attention mechanisms.

## 1 Introduction

A rapidly growing class of model architectures for graph representation learning are Graph Neural Networks (GNNs) [20] which have achieved strong empirical performance across various applications such as social network analysis [8], drug discovery [27], recommendation systems [61], and traffic forecasting [50]. This has driven research focused on constructing and assessing specific architectural designs [59] tailored to various tasks.

On the theoretical front, mostly the expressive power [51, 26] and computational capabilities [45] of GNNs have been studied. However, there is limited understanding of the underlying learning mechanics. We undertake a classical variant of GNNs, the Graph Attention Network (GAT) [48, 9]. It overcomes the limitation of standard neighborhood representation averaging in GCNs [29, 22] by employing a self-attention mechanism [47] on nodes. Attention performs a weighted aggregation over a node's neighbors to learn their relative importance. This increases the expressiveness of the neural network. GAT is a popular model that continues to serve as strong baseline.

However, a major limitation of message-passing GNNs in general, and therefore GATs as well, is severe performance degradation when the depth of the network is even slightly increased. SOTA results are reportedly achieved by models of 2 or 3 layers. This problem has largely been attributed to over-smoothing [35], a phenomenon in which node representations become indistinguishable when multiple layers are stacked. To relieve GNNs of over-smoothing, practical techniques inspired by classical deep neural networks tweak the training process, e.g. by normalization [11, 64, 65, 66] or regularization [38, 42, 55, 67], and impose architectural changes such as skip connections (dense

37th Conference on Neural Information Processing Systems (NeurIPS 2023).

and residual) [34, 53] or offer other engineering solutions [33] or their combinations [12]. Other approaches to overcome over-smoothing include learning CNN-like spatial operators from random paths in the graph instead of the point-wise graph Laplacian operator [17] and learning adaptive receptive fields (different 'effective' neighborhoods for different nodes) [63, 36, 52]. Another problem associated with loss of performance in deeper networks is over-squashing [1], but we do not discuss it since non-attentive models are more susceptible to it, which is not our focus.

Alternative causes for the challenged trainability of deeper GNNs beyond over-smoothing is hampered signal propagation during training (i.e. backpropagation) [25, 62, 13]. To alleviate this problem for GCNs, a dynamic addition of skip connections to vanilla-GCNs, which is guided by gradient flow, and a topology-aware isometric initialization are proposed in [25]. A combination of orthogonal initialization and regularization facilitates gradient flow in [21]. To the best of our knowledge, these are the only works that discuss the trainability issue of deep GNNs from the perspective of signal propagation, which studies a randomly initialized network at initialization. Here, we offer an insight into a mechanism that allows these approaches to improve the trainability of deeper networks that is related to the entire gradient flow dynamics. We focus in particular on GAT architectures and the specific challenges that are introduced by attention.

Our work translates the concept of neuronwise balancedness from traditional deep neural networks to GNNs, where a deeper understanding of gradient dynamics has been developed and norm balancedness is a critical assumption that induces successful training conditions. One reason is that the degree of initial balancedness is conserved throughout the training dynamics as described by gradient flow (i.e. a model of gradient descent with infinitesimally small learning rate or step size). Concretely, for fully connected feed-forward networks and deep convolutional neural networks with continuous homogenous activation functions such as ReLUs, the difference between the squared $l2$-norms of incoming and outgoing parameters to a neuron stays approximately constant during training [15, 32]. Realistic optimization elements such as finite learning rates, batch stochasticity, momentum, and weight decay break the symmetries induced by these laws. Yet, gradient flow equations can be extended to take these practicalities into account [31].

Inspired by these advances, we derive a conservation law for GATs with positive homogeneous activation functions such as ReLUs. As GATs are a generalization of GCNs, most aspects of our insights can also be transferred to other architectures. Our consideration of the attention mechanism would also be novel in the context of classic feed-forward architectures and could have intriguing implications for transformers that are of independent interest. In this work, however, we focus on GATs and derive a relationship between model parameters and their gradients, which induces a conservation law under gradient flow. Based on this insight, we identify a reason for the lack of trainability in deeper GATs and GCNs, which can be alleviated with a balanced parameter initialization. Experiments on multiple benchmark datasets demonstrate that our proposal is effective in mitigating the highlighted trainability issues, as it leads to considerable training speed-ups and enables significant parameter changes across all layers. This establishes a causal relationship between trainability and parameter balancedness also empirically, as our theory predicts. The main theorem presented in this work serves as a stepping stone to studying the learning dynamics of positive homogeneous models with attention mechanisms such as transformers [47] and in particular, vision transformers which require depth more than NLP transformers[49]. Our contributions are as follows.

- We derive a conservation law of gradient flow dynamics for GATs, including its variations such as shared feature weights and multiple attention heads.

- This law offers an explanation for the lack of trainability of deeper GATs with standard initialization.

- To demonstrate empirically that our theory has established a causal link between balancedness and trainability, we propose a balanced initialization scheme that improves the trainability of deeper networks and attains considerable speedup in training and convergence time, as our theory predicts.

## 2 Theory: Conservation laws in GATs

**Preliminaries** For a graph $G = (V, E)$ with node set $V$ and edge set $E \subseteq V \times V$, where the neighborhood of a node $v$ is given as $\mathcal{N}(v) = \{u|(u,v) \in E\}$, a GNN layer computes each node's representation by aggregating over its neighbors' representation. In GATs, this aggregation is weighted by parameterized attention coefficients $\alpha_{uv}$, which indicate the importance of node $u$ for $v$.

Table 1: Attributes of init. schemes

| Init. | Feat. | Attn. | Bal.? |
|---|---|---|---|
| Xav | Xavier | Xavier | No |
| $\text{Xav}_Z$ | Xavier | Zero | No |
| $\text{Bal}_X$ | Xavier | Zero | Yes |
| $\text{Bal}_O$ | LLortho | Zero | Yes |

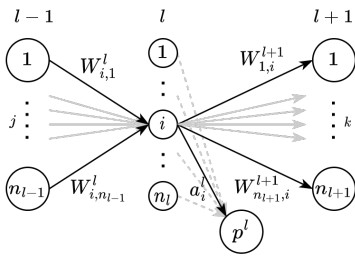

Figure 1: Params. to balance for neuron $i \in [n_l]$

Given input representations $h_v^{l-1}$ for all nodes $v \in V$, a GAT [1] layer $l$ transforms those to:

$$h_v^l = \phi(\sum_{u \in \mathcal{N}(v)} \alpha_{uv}^l \cdot W_s^l h_u^{l-1}), \quad \text{where} \tag{1}$$

$$\alpha_{uv}^l = \frac{\exp((a^l)^\top \cdot \text{LeakyReLU}(W_s^l h_u^{l-1} + W_t^l h_v^{l-1}))}{\sum_{u' \in \mathcal{N}(v)} \exp((a^l)^\top \cdot \text{LeakyReLU}(W_s^l h_{u'}^{l-1} + W_t^l h_v^{l-1}))}. \tag{2}$$

We consider $\phi$ to be a positively homogeneous activation functions (i.e $\phi(x) = x\phi'(x)$ and consequently, $\phi(ax) = a\phi(x)$ for positive scalars $a$), such as a ReLU $\phi(x) = \max\{x, 0\}$ or LeakyReLU $\phi(x) = \max\{x, 0\} + -\alpha \max\{-x, 0\}$. The feature transformation weights $W_s$ and $W_t$ for source and target nodes, respectively, may also be shared such that $W = W_s = W_t$.

**Definition 2.1.** Given training data $\{(x_m, y_m)\}_{m=1}^M \subset \mathbb{R}^d \times \mathbb{R}^p$ for $M \leq V$, let $f : \mathbb{R}^d \to \mathbb{R}^p$ be the function represented by a network constructed by stacking $L$ GAT layers as defined in Eq. (1) and (2) with $W = W_s = W_t$ and $h_m^0 = g(x_m)$. Each layer $l \in [L]$ of size $n_l$ has associated parameters: a feature weight matrix $W^l \in \mathbb{R}^{n_l \times n_{l-1}}$ and an attention vector $a^l \in \mathbb{R}^{n_l}$, where $n_0 = d$ and $n_L = p$. Given a differentiable loss function $\ell : \mathbb{R}^d \times \mathbb{R}^p \to \mathbb{R}$, the loss $\mathcal{L} = 1/M \sum_{i=1}^M \ell(f(x_m), y_m)$ is used to update model parameters $w \in \{W^l, a^l\}_{l=1}^L$ with learning rate $\gamma$ by gradient descent, i.e., $w^{t+1} = w^t - \gamma \nabla_w \mathcal{L}$, where $\nabla_w \mathcal{L} = [\partial \mathcal{L}/\partial w_1, \dots, \partial \mathcal{L}/\partial w_{|w|}]$ and $w^0$ is set by the initialization scheme. For an infinitesimal $\gamma \to 0$, the dynamics of gradient descent behave similarly to gradient flow defined by $dw/dt = -\nabla_w \mathcal{L}$, where $t$ is the continuous time index.

We use $W[i, :]$, $W[:, i]$, and $a[i]$ to denote the $i^{th}$ row of $W$, column of $W$, and entry of $a$, respectively. Note that $W^l[i, :]$ is the vector of weights incoming to neuron $i \in [n_l]$ and $W^{l+1}[:, i]$ is the vector of weights outgoing from the same neuron. For the purpose of discussion, let $i \in [d_l], j \in [d_{l-1}]$, and $k \in [d_{l+1}]$, as depicted in Fig. 1. $\langle \cdot, \cdot \rangle$ denotes the scalar product.

**Conservation Laws**  We focus the following exposition on weight-sharing GATs, as this variant has the practical benefit that it requires fewer parameters. Yet, similar laws also hold for the non-weight-sharing case, as we state and discuss in the supplement. For brevity, we also defer the discussion of laws for the multi-headed attention mechanism to the supplement.

**Theorem 2.2** (Structure of gradients). *Given the feature weight and attention parameters $W^l$ and $a^l$ of a layer $l$ in a GAT network as described in Def. 2.1, the structure of the gradients for layer $l \in [L-1]$ in the network is conserved according to the following law:*

$$\langle W^l[i, :], \nabla_{W^l[i, :]} \mathcal{L} \rangle - \langle a^l[i], \nabla_{a^l[i]} \mathcal{L} \rangle = \langle W^{l+1}[:, i], \nabla_{W^{l+1}[:, i]} \mathcal{L} \rangle. \tag{3}$$

**Corollary 2.3** (Norm conservation of weights incoming and outgoing at every neuron). *Given the gradient structure of Theorem 2.2 and positive homogeneity of the activation $\phi$ in Eq. (1), gradient flow in the network adheres to the following invariance for $l \in [L-1]$:*

$$\frac{d}{dt}\left(\left\|W^l[i, :]\right\|^2 - \left\|a^l[i]\right\|^2\right) = \frac{d}{dt}\left(\left\|W^{l+1}[:, i]\right\|^2\right), \tag{4}$$

---

[1]This definition of GAT is termed GATv2 in [9]. Hereafter, we refer to GATv2 as GAT.

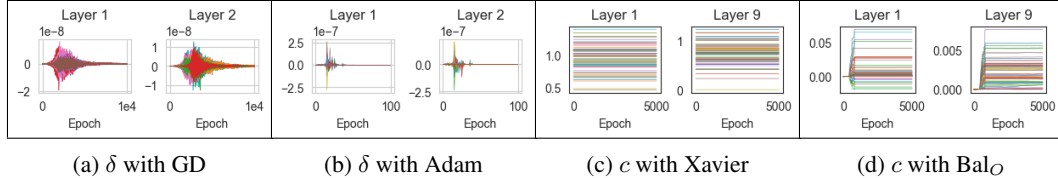

(a) $\delta$ with GD      (b) $\delta$ with Adam      (c) $c$ with Xavier      (d) $c$ with $\text{Bal}_O$

Figure 2: Visualizing conservation laws: (a),(b) show $\delta \approx 0$ (can not empirically be exactly zero due to finite $\gamma = 0.1$) from Eq. (7) for hidden layers when $L = 3$. (c),(d) show the value of $c$ from Eq. (5), which is determined by the initialization and should be 0 when the network is balanced, for $L = 10$ trained with GD. With Xav., the network is unbalanced and hence $c \neq 0$ for all neurons. With $\text{Bal}_O$, $c = 0$ exactly at initialization for all neurons by design but during training, it varies slightly and $c \approx 0$ as the network becomes slightly unbalanced due to the finite $\gamma = 0.1$. As both $\delta$ and $c$ approximately meet their required value, we conclude that the derived law holds for practical purposes as well.

*It then directly follows by integration over time that the l2-norms of feature and attention weights incoming to and outgoing from a neuron are preserved such that*

$$\left\|W^l[i,:]\right\|^2 - \left\|a^l[i]\right\|^2 - \left\|W^{l+1}[:i]\right\|^2 = c, \tag{5}$$

*where $c = \dfrac{\mathrm{d}}{\mathrm{d}t}\left(\left\|W^l[i,:]\right\|^2 - \left\|a^l[i]\right\|^2\right)$ at initialization (i.e. $t = 0$).*

We denote the 'degree of balancedness' by $c$ and call an initialization balanced if $c = 0$.

**Corollary 2.4** (Norm conservation across layers). *Given Eq. (4), the invariance of gradient flow at the layer level for $l \in [L-1]$ by summing over $i \in [n_l]$ is:*

$$\frac{\mathrm{d}}{\mathrm{d}t}\left(\left\|W^l\right\|_F^2 - \left\|a^l\right\|^2\right) = \frac{\mathrm{d}}{\mathrm{d}t}\left(\left\|W^{l+1}\right\|_F^2\right). \tag{6}$$

*Remark* 2.5. Similar conservation laws also hold for the original less expressive GAT version [48] as well as for the vanilla GCN [29] yielded by fixing $a^l = \mathbf{0}$.

**Insights** We first verify our theory and then discuss its implications in the context of four different initializations for GATs, which are summarized in Table 1. Xavier (Xav.) initialization [19] is the standard default for GATs. We describe the Looks-Linear-Orthogonal (LLortho) initialization later.

In order to empirically validate Theorem 2.2, we rewrite Equation 3 to define $\delta$ as:

$$\delta = \langle W^l[i,:], \nabla_{W^l[i,:]}\mathcal{L}\rangle - \langle a^l, \nabla_{a^l}\mathcal{L}\rangle - \langle W^{l+1}[:,i], \nabla_{W^{l+1}[:,i]}\mathcal{L}\rangle = 0. \tag{7}$$

We observe how the value of $\delta$ varies during training in Fig. 2 for both GD and Adam optimizers with $\gamma = 0.1$. Although the theory assumes infinitesimal learning rates due to gradient flow, it still holds sufficiently ($\delta \approx 0$) in practice for normally used learning rates as we validate in Fig. 2. Furthermore, although the theory assumes vanilla gradient descent optimization, it still holds for the Adam optimizer (Fig. 2b). This is so because the derived law on the level of scalar products between gradients and parameters (see Theorem 2.2) still holds, as it states a relationship between gradients and parameters that is independent of the learning rate or precise gradient updates.

Having verified Theorem 2.2, we use it to deduce an explanation for a major trainability issue of GATs with Xavier initialization that is amplified with increased network depth.

The main reason is that the last layer has a comparatively small average weight norm, as $\mathbb{E}\left\|W^L[:,i]\right\|^2 = 2n_L/(n_L + n_{L-1}) \ll 1$, where the number of outputs is smaller than the layer width $n_L \ll n_{L-1}$. In contrast, $\mathbb{E}\left\|W^{L-1}[i,:]\right\|^2 = 2n_{L-1}/(n_{L-1} + n_{L-2}) = 1$ and $\mathbb{E}a^{L-1}[i]^2 = 2/(1 + n_{L-1})$. In consequence, the right-hand side of our derived conservation law

$$\sum_{j=1}^{n_{L-2}} (W_{ij}^{L-1})^2 \frac{\nabla_{W_{ij}^{L-1}}\mathcal{L}}{W_{ij}^{L-1}} - (a_i^{L-1})^2 \frac{\nabla_{a_i^{L-1}}\mathcal{L}}{a_i^{L-1}} = \sum_{k=1}^{n_L} (W_{ki}^L)^2 \frac{\nabla_{W_{ki}^L}\mathcal{L}}{W_{ki}^L} \tag{8}$$

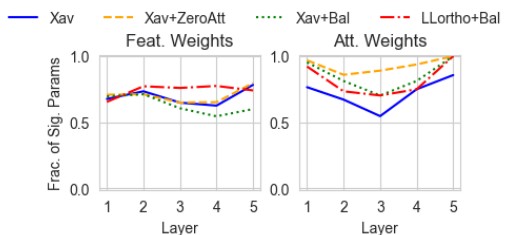
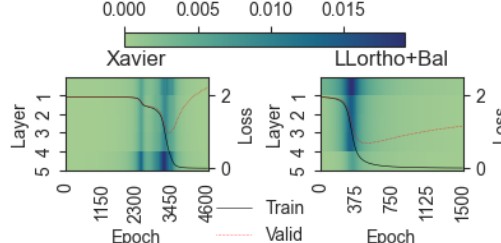

(a) Frac. of sig. params with relative change $> 0.05$    (b) Rel. grad. norms of feat. wghts. & training curve

Figure 3: For $L = 5$, a high fraction of parameters change, and gradients propagate to earlier layers for both the unbalanced and balanced initialization Xav. and $\text{Bal}_O$ initialization achieve $75.5\%$ and $79.9\%$ test accuracy, respectively. Note that the balanced initialization in (b) is able to train faster.

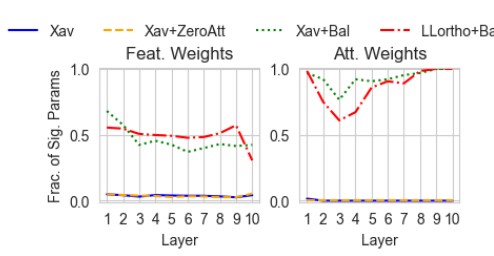
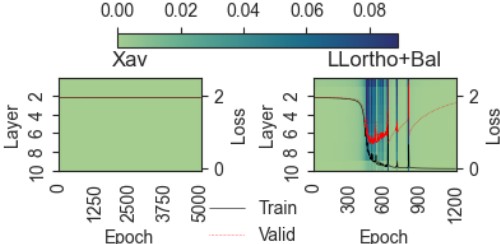

(a) Frac. of sig. params with relative change $> 0.5$    (b) Rel. grad. norms of feat. wghts. & training curve

Figure 4: For $L = 10$, unbalanced initialization is unable to train the model as relative gradients are extremely small, and the parameters (and loss) do not change. However, the balanced initialization is able to propagate gradients, change parameters, and thus attain training loss $\rightarrow 0$. Xav. and $\text{Bal}_O$ achieve $39.3\%$ and $80.2\%$ test accuracy, respectively. Note that the balanced initialization in (b) is also able to train faster than the 5 layer network with Xavier initialization in Fig. 3b.

is relatively small, as $\sum_{k=1}^{n_L} (W_{ki}^L)^2 << 1$, on average, while the weights on the left-hand side are orders of magnitude larger. This implies that the relative gradients on the left-hand side are likely relatively small to meet the equality unless all the signs of change are set in such a way that they satisfy the equality, which is highly unlikely during training. Therefore, relative gradients in layer $L-1$ and accordingly also the previous layers of the same dimension can only change in minor ways.

The amplification of this trainability issue with depth can also be explained by the recursive substitution of Theorem 2.2 in Eq. (8) that results in a telescoping series yielding:

$$\sum_{j=1}^{n_1} \sum_{m=1}^{n_0} W_{jm}^{(1)^2} \frac{\nabla_{W_{jm}^{(1)}} \mathcal{L}}{W_{jm}^{(1)}} - \sum_{l=1}^{L-1} \sum_{o=1}^{n_l} a_o^{(l)^2} \frac{\nabla_{a_o^{(l)}} \mathcal{L}}{a_o^{(l)}} = \sum_{i=1}^{n_{L-1}} \sum_{k=1}^{n_L} W_{ki}^{(L)^2} \frac{\nabla W_{ki}^{(L)} \mathcal{L}}{W_{ki}^{(L)}} \qquad (9)$$

Generally, $2n_1 < n_0$ and thus $\mathbb{E} \left\| W^1[j:] \right\|^2 = 2n_1/(n_1 + n_0) < 1$. Since the weights in the first layer and the gradients propagated to the first layer are both small, gradients of attention parameters of the intermediate hidden layers must also be very small in order to satisfy Eq. (9). Evidently, the problem aggravates with depth where the same value must now be distributed over the parameters and gradients of more layers. Note that the norms of the first and last layers are usually not arbitrary but rather determined by the input and output scale of the network.

Analogously, we can see see how a balanced initialization would mitigate the problem. Equal weight norms $\left\| W^{L-1}[i, :] \right\|^2$ and $\left\| W^L[:, i] \right\|^2$ in Eq. (8) (as the attention parameters are set to $0$ during balancing, see Procedure 2.6) would allow larger relative gradients in layer $L-1$, as compared to the imbalanced case, that can enhance gradient flow in the network to earlier layers. In other words, gradients on both sides of the equation have equal room to drive parameter change.

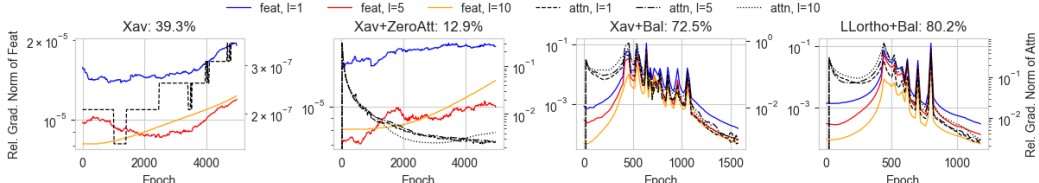

Figure 5: Relative gradient norms of feature (left axis, solid) and of attention (right axis, stylized) parameters for $l \in [1, 5, 10]$ and $L = 10$, sampled every 25 epochs. Test accuracy is at the top. Both attention and feature gradients at the first, middle, and last layer of the network with both balanced initializations are much larger than with unbalanced initialization (note axis scales).

To visualize the trainability issue, we study the relative change $|(w^* - w_0)/w^*|$. of trained network parameters $(w^*)$ w.r.t. their initial value. In order to observe meaningful relative change, we only consider parameters with a significant contribution to the model output $(w^* \geq 10^{-4})$. We also plot the relative gradient norms of the feature weights across all layers to visualize their propagation. We display these values for $L = 5$ and $L = 10$ in Fig. 3 and 4, respectively, and observe a stark contrast, similar to the trends in relative gradient norms of selected layers of a 10 layer GAT in Fig. 5. In all experiments, GAT was trained with the specified initialization on Cora using GD with $\gamma = 0.1$ for 5000 epochs.

Interestingly, the attention parameters change most in the first layer and increasingly more towards the last layer, as seen in Fig. 3a and 4a. This is consistently observed in both the 5 and 10 layer networks if the initialization is balanced, but with unbalanced initialization, the 10 layer network is unable to produce the same effect. As our theory defines a coarse-level conservation law, such fine-grained training dynamics cannot be completely explained by it.

Since the attention parameters are set to $0$ in the balanced initializations (see Procedure 2.6), as ablation, we also observe the effect of initializing attention and feature parameters with zero and Xavier, respectively. From Eq.(2), $a^l = 0$ leads to $\alpha_{uv} = 1/|\mathcal{N}(v)|$, implying that all neighbors $u$ of the node $v$ are equally important at initialization. Intuitively, this allows the network to learn the importance of neighbors without any initially introduced bias and thus avoids the 'chicken and egg' problem that arises in the initialization of attention over nodes[30]. Although this helps generalization in some cases (see Fig. 6), it alone does not help the trainability issue as seen in Fig. 4a. It may in fact worsen it (see Fig. 5), which is explained by Eq. 8. These observations underline the further need for parameter balancing.

**Procedure 2.6** (Balancing). Based on Eq. (5) from the norm preservation law 2.3, we note that in order to achieve balancedness, (i.e. set $c = 0$ in Eq.(5)), the randomly initialized parameters $W^l$ and $a^l$ must satisfy the following equality for $l \in [L]$:

$$\left\| W^l[i, :] \right\|^2 - \left\| a^l[i] \right\|^2 = \left\| W^{l+1}[:, i] \right\|^2$$

This can be achieved by scaling the randomly initialized weights as follows:

1. Set $a^l = \mathbf{0}$ for $l \in [L]$.

2. Set $W^1[i, :] = \frac{W^1[i, :]}{\|W^1[i, :]\|} \sqrt{\beta_i}$, for $i \in [n_1]$ where $\beta_i$ is a hyperparameter

3. Set $W^{l+1}[:, i] = \frac{W^{l+1}[:, i]}{\|W^{l+1}[:, i]\|} \left\| W^l[i, :] \right\|$ for $i \in [n_l]$ and $l \in [L-1]$

In step 2, $\beta_i$ determines the initial squared norm of the incoming weights to neuron $i$ in the first layer. It can be any constant thus also randomly sampled from a normal or uniform distribution. We explain why we set $\beta_i = 2$ for $i \in [n_1]$ in the context of an orthogonal initialization.

*Remark* 2.7. This procedure balances the network starting from $l = 1$ towards $l = L$. Yet, it could also be carried out in reverse order by first defining $\left\| W^L[:, i] \right\|^2 = \beta_i$ for $i \in [n_{L-1}]$.

**Balanced Orthogonal Initialization**  The feature weights $W^l$ for $l \in [L]$ need to be initialized randomly before balancing, which can simply be done by using Xavier initialization. However, existing works have pointed out benefits of orthogonal initialization to achieve initial dynamical

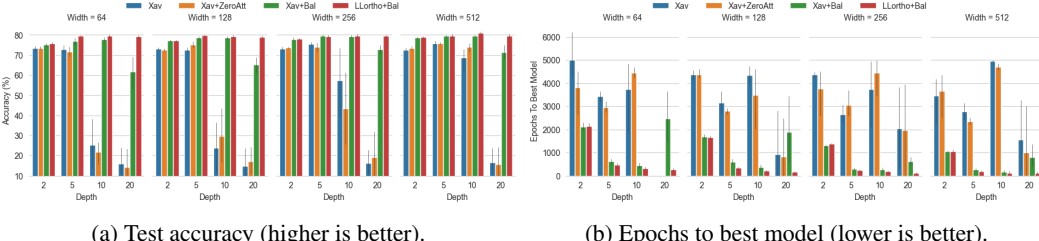

(a) Test accuracy (higher is better).    (b) Epochs to best model (lower is better).

Figure 6: GAT trained on Cora using SGD

isometry for DNNs and CNNs [37, 44, 23], as it enables training very deep architectures. In line with these findings, we initialize $W^l$, before balancing, to be an orthogonal matrix with a looks-linear (LL) mirrored block structure [46, 6], that achieves perfect dynamical isometry in perceptrons with ReLU activation functions. Due to the peculiarity of neighborhood aggregation, the same technique does not induce perfect dynamical isometry in GATs (or GNNs). Exploring how dynamical isometry can be achieved or approximated in general GNNs could be an interesting direction for future work. Nevertheless, using a (balanced) LL-orthogonal initialization enhances trainaibility, particularly of deeper models. We defer further discussion on the effects of orthogonal initialization, such as with identity matrices, to the appendix.

We outline the initialization procedure as follows: Let $\|$ and $\models$ denote row-wise and column-wise matrix concatenation, respectively. Let us draw a random orthonormal submatrix $U^l$ and define $W^1 = [U^1 \| -U^1]$ where $U^1 \in \mathbb{R}^{\frac{n_1}{2} \times n_0}$, $W^l = [[U^l \| -U^l] \models [-U^l \| U^l]]$ where $U^l \in \mathbb{R}^{\frac{n_l}{2} \times \frac{n_{l-1}}{2}}$ for $l = \{2, \ldots, L-1\}$, and $W^L = [U^L \models -U^L]$ where $U^L \in \mathbb{R}^{n_L \times \frac{n_{l-1}}{2}}$. Since $U^L$ is an orthonormal matrix, $\left\|W^L[:, i]\right\|^2 = 2$ by definition, and by recursive application of Eq. (2.3) (with $a^l = \mathbf{0}$), balancedness requires that $\left\|W^1[i :,]\right\|^2 = 2$. Therefore, we set $\beta_i = 2$ for $i \in n_1$.

## 3  Experiments

The main purpose of our experiments is to verify the validity of our theoretical insights and deduce an explanation for a major trainability issue of GATs that is amplified with increased network depth. Based on our theory, we understand the striking observation in Figure 4a that the parameters of default Xavier initialized GATs struggle to change during training. We demonstrate the validity of our theory with the nature of our solution. As we balance the parameter initialization (see Procedure 2.6), we allow the gradient signal to pass through the network layers, which enables parameter changes during training in all GAT layers. In consequence, balanced initialization schemes achieve higher generalization performance and significant training speed ups in comparison with the default Xavier initialization. It also facilitates training deeper GAT models.

To analyze this effect systematically, we study the different GAT initializations listed in Table 1 on generalization (in % accuracy) and training speedup (in epochs) using nine common benchmark datasets for semi-supervised node classification tasks. We defer dataset details to the supplement. We use the standard provided train/validation/test splits and have removed the isolated nodes from Citeseer. We use the Pytorch Geometric framework and run our experiments on either Nvidia T4 Tensor Core GPU with 15 GB RAM or Nvidia GeForce RTX 3060 Laptop GPU with 6 GB RAM. We allow each network to train, both with SGD and Adam, for 5000 epochs (unless it converges earlier, i.e. achieves training loss $\leq 10^{-4}$) and select the model state with the highest validation accuracy. For each experiment, the mean $\pm 95\%$ confidence interval over five runs is reported for accuracy and epochs to the best model. All reported results use ReLU activation, weight sharing and no biases, unless stated otherwise. No weight sharing leads to similar trends, which we therefore omit. Our experimental code is available at `https://github.com/RelationalML/GAT_Balanced_Initialization`.

**SGD** We first evaluate the performance of the different initialization schemes of GAT models using vanilla gradient descent. Since no results have been previously reported using SGD on these datasets, to the best of our knowledge, we find that setting a learning rate of 0.1, 0.05 and 0.005 for $L = [2, 5]$, $L = [10, 20]$, and $L = 40$, respectively, allows for reasonably stable training on Cora, Citeseer, and

Table 2: Mean accuracy(%) $\pm$ 95% CI over five runs of GAT with width= 64 trained using SGD.

| | Init. | $L = 2$ | $L = 5$ | $L = 10$ | $L = 20$ | $L = 40$ |
|---|---|---|---|---|---|---|
| **Citeseer** | Xav | $71.82 \pm 2.73$ | $58.40 \pm 2.25$ | $24.70 \pm 8.90$ | $19.23 \pm 1.54$ | $18.72 \pm 1.15$ |
| | $\mathrm{Bal}_X$ | $71.62 \pm 0.80$ | $68.83 \pm 1.62$ | $64.13 \pm 1.57$ | $54.88 \pm 7.95$ | $42.63 \pm 17.47$ |
| | $\mathrm{Bal}_O$ | $\mathbf{72.02 \pm 0.63}$ | $\mathbf{70.63 \pm 0.60}$ | $\mathbf{68.83 \pm 1.68}$ | $\mathbf{68.70 \pm 1.18}$ | $\mathbf{63.40 \pm 1.43}$ |
| **Pubmed** | Xav | $77.26 \pm 1.39$ | $70.68 \pm 2.16$ | $67.32 \pm 10.70$ | $36.52 \pm 11.50$ | $27.20 \pm 13.99$ |
| | $\mathrm{Bal}_X$ | $\mathbf{78.02 \pm 0.73}$ | $75.66 \pm 1.81$ | $\mathbf{77.60 \pm 1.56}$ | $76.44 \pm 1.70$ | $75.74 \pm 2.94$ |
| | $\mathrm{Bal}_O$ | $77.68 \pm 0.45$ | $\mathbf{76.62 \pm 1.59}$ | $77.04 \pm 2.14$ | $\mathbf{78.20 \pm 0.61}$ | $\mathbf{77.80 \pm 1.41}$ |
| **Actor** | Xav | $27.32 \pm 0.59$ | $24.60 \pm 0.93$ | $24.08 \pm 0.80$ | $22.29 \pm 3.26$ | $19.46 \pm 5.75$ |
| | $\mathrm{Bal}_X$ | $26.00 \pm 0.59$ | $23.93 \pm 1.42$ | $\mathbf{24.21 \pm 0.78}$ | $\mathbf{24.74 \pm 1.14}$ | $23.88 \pm 0.97$ |
| | $\mathrm{Bal}_O$ | $\mathbf{26.59 \pm 1.03}$ | $\mathbf{24.61 \pm 0.77}$ | $24.17 \pm 0.62$ | $24.24 \pm 1.05$ | $\mathbf{23.93 \pm 1.53}$ |
| **Chameleon** | Xav | $\mathbf{52.81 \pm 1.37}$ | $54.21 \pm 1.05$ | $30.31 \pm 5.96$ | $22.19 \pm 2.04$ | $22.28 \pm 3.15$ |
| | $\mathrm{Bal}_X$ | $51.18 \pm 1.94$ | $\mathbf{54.21 \pm 0.82}$ | $\mathbf{52.11 \pm 3.72}$ | $51.89 \pm 1.89$ | $38.64 \pm 10.31$ |
| | $\mathrm{Bal}_O$ | $50.00 \pm 3.07$ | $53.95 \pm 1.81$ | $51.84 \pm 3.21$ | $\mathbf{52.72 \pm 0.13}$ | $\mathbf{44.30 \pm 1.61}$ |
| **Cornell** | Xav | $\mathbf{42.70 \pm 2.51}$ | $41.08 \pm 2.51$ | $\mathbf{42.70 \pm 1.34}$ | $25.41 \pm 14.64$ | $22.70 \pm 13.69$ |
| | $\mathrm{Bal}_X$ | $41.08 \pm 6.84$ | $35.14 \pm 11.82$ | $41.08 \pm 2.51$ | $40.00 \pm 4.93$ | $\mathbf{37.84 \pm 5.62}$ |
| | $\mathrm{Bal}_O$ | $42.16 \pm 1.64$ | $\mathbf{43.24 \pm 2.12}$ | $36.76 \pm 5.02$ | $\mathbf{35.68 \pm 3.29}$ | $36.22 \pm 3.42$ |
| **Squirrel** | Xav | $35.20 \pm 0.44$ | $40.96 \pm 0.92$ | $21.65 \pm 1.52$ | $20.23 \pm 1.69$ | $19.67 \pm 0.29$ |
| | $\mathrm{Bal}_X$ | $\mathbf{35.95 \pm 1.69}$ | $40.98 \pm 0.87$ | $\mathbf{38.98 \pm 1.49}$ | $38.35 \pm 1.07$ | $25.38 \pm 4.62$ |
| | $\mathrm{Bal}_O$ | $35.83 \pm 0.92$ | $\mathbf{42.52 \pm 1.19}$ | $38.85 \pm 1.36$ | $\mathbf{39.15 \pm 0.44}$ | $\mathbf{26.57 \pm 1.83}$ |
| **Texas** | Xav | $60.00 \pm 1.34$ | $60.54 \pm 3.42$ | $58.92 \pm 1.34$ | $49.73 \pm 20.97$ | $17.84 \pm 26.98$ |
| | $\mathrm{Bal}_X$ | $\mathbf{60.54 \pm 1.64}$ | $\mathbf{61.62 \pm 2.51}$ | $\mathbf{61.62 \pm 2.51}$ | $\mathbf{58.92 \pm 2.51}$ | $56.22 \pm 1.34$ |
| | $\mathrm{Bal}_O$ | $60.00 \pm 1.34$ | $57.30 \pm 1.34$ | $56.76 \pm 0.00$ | $58.38 \pm 4.55$ | $\mathbf{57.30 \pm 3.91}$ |
| **Wisconsin** | Xav | $\mathbf{51.37 \pm 6.04}$ | $51.37 \pm 8.90$ | $51.76 \pm 3.64$ | $43.14 \pm 25.07$ | $31.76 \pm 31.50$ |
| | $\mathrm{Bal}_X$ | $49.80 \pm 8.79$ | $\mathbf{54.51 \pm 4.19}$ | $47.84 \pm 7.16$ | $\mathbf{50.59 \pm 10.49}$ | $41.18 \pm 1.54$ |
| | $\mathrm{Bal}_O$ | $49.80 \pm 4.24$ | $55.69 \pm 3.64$ | $\mathbf{51.76 \pm 5.68}$ | $49.02 \pm 4.36$ | $\mathbf{48.24 \pm 4.52}$ |

Pubmed. For the remaining datasets, we set the learning rate to $0.05, 0.01, 0.005$ and $0.0005$ for $L = [2, 5], L = 10, L = 20$, and $L = 40$, respectively. Note that we do not perform fine-tuning of the learning rate or other hyperparameters with respect to performance and use the same settings for all initializations to allow fair comparison.

Figure 6 highlights that models initialized with balanced parameters, $\mathrm{Bal}_O$, consistently achieve the best accuracy and significantly speed up training, even on shallow models of 2 layers. In line with our theory, the default Xavier (Xav) initialization hampers model training (see also Figure 4a). Interestingly, the default Xavier (Xav) initialized deeper models tend to improve in performance with width but cannot compete with balanced initialization schemes. For example, the accuracy achieved by Xav for $L = 10$ increases from $24.7\%$ to $68.7\%$ when the width is increased from $64$ to $512$. We hypothesize that the reason why width aids generalization performance is that a higher number of (almost random) features supports better models. This would also be in line with studies of overparameterization in vanilla feed-forward architectures, where higher width also aids random feature models [2, 4, 57, 43].

The observation that width overparameterized models enable training deeper models may be of independent interest in the context of how overparameterization in GNNs may be helpful. However, training wider *and* deeper (hence overall larger) models is computationally inefficient, even for datasets with the magnitude of Cora. In contrast, the $\mathrm{Bal}_O$ initialized model for $L = 10$ is already able to attain $79.7\%$ even with width= 64 and improves to $80.9\%$ with width= 512. Primarily the

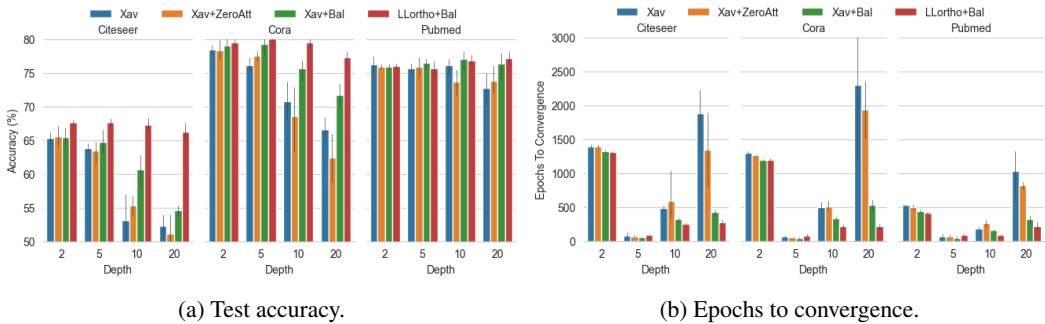

(a) Test accuracy.

(b) Epochs to convergence.

Figure 7: GAT with $64$ hidden dimensions trained using Adam.

parameter balancing must be responsible for the improved performance, as it is not sufficient to simply initialize the attention weights to zero, as shown in Figure 4a and 5.

For the remaining datasets, we report only the performance of networks with $64$ hidden dimension in Table 2 for brevity. Since we have considered the $\text{Xav}_Z$ initialization only as an ablation study and find it to be ineffective (see Figure 6a), we do not discuss it any further.

Note that for some datasets (e.g.Pubmed, Wisconsin), deeper models (e.g. $L = 40$) are unable to train at all with Xavier initialization, which explains the high variation in test accuracy across multiple runs as, in each run, the model essentially remains a random network. The reduced performance even with balanced initialization at higher depth may be due to a lack of convergence within $5000$ epochs. Nevertheless, the fact that the drop in performance is not drastic, but rather gradual as depth increases, indicates that the network is being trained, i.e. trainability is not an issue.

As a stress test, we validate that GAT networks with balanced initialization maintain their trainability at even larger depths of 64 and 80 layers (see Fig. 9 in appendix). In fact, the improved performance and training speed-up of models with balanced initialization as opposed to models with standard initialization is upheld even more so for very deep networks.

Apart from Cora, Citeseer and Pubmed, the other datasets are considered heterophilic in nature and standard GNN models do not perform very well on them. State-of-the-art performance on these datasets is achieved by specialized models. We do not compare with them for two reasons: i) they do not employ an attention mechanism, which is the focus of our investigations, but comprise various special architectural elements and ii) our aim is not to outperform the SOTA, but rather highlight a mechanism that underlies the learning dynamics and can be exploited to improve trainability. We also demonstrate the effectiveness of a balanced initialization in training deep GATs in comparison to LipschitzNorm [14], a normalization technique proposed specifically for training deep GNNs with self-attention layers such as GAT (see Table 6 in appendix).

**Adam**    Adam, the most commonly used optimizer for GNNs, stabilizes the training dynamics and can compensate partially for problematic initializations. However, the drop in performance with depth, though smaller, is inevitable with unbalanced initialization. As evident from Figure 7, $\text{Bal}_O$ initialized models achieve higher accuracy than or at par with Xavier initialization and converge in fewer epochs. As we observe with SGD, this difference becomes more prominent with depth, despite the fact that Adam itself significantly improves the trainability of deeper networks over SGD. Our argument of small relative gradients (see Eq.(8)) also applies to Adam. We have used the initial learning rates reported in the literature [9]: $0.005$ for Cora and Citeseer, and $0.01$ for Pubmed for the $2$ and $5$ layer networks. To allow stable training of deeper networks, we reduce the initial learning rate by a factor $0.1$ for the $10$ and $20$ layer networks on all three datasets.

**Architectural variations**    We also consider other architectural variations such as employing ELUs instead of ReLUs, using multiple attention heads, turning off weight sharing, and addition of standard elements such as weight decay and dropout. In all cases, the results follow a similar tendency as already reported. Due to a lack of space, we defer the exact results to the appendix. These variations further increase the performance of networks with our initialization proposals, which can therefore be regarded as complementary. Note that residual skip connections between layers are also supported

in a balanced initialization provided their parameters are initialized with zero. However, to isolate the contribution of the initialization scheme and validate our theoretical insights, we have focused our exposition on the vanilla GAT version.

**Limitations**   The derived conservation law only applies to the self-attention defined in the original GAT and GATv2 models, and their architectural variations such as $\omega$GAT[18] (see Fig. 10 in appendix). Note that the law also holds for the non-attentive GCNs (see Table 7 in appendix) which are a special case of GATs (where the attention parameters are simply zero). Modeling different kinds of self-attention such as the dot-product self-attention in [28] entails modification of the conservation law, which has been left for future work.

## 4   Discussion

GATs [9] are powerful graph neural network models that form a cornerstone of learning from graph-based data. The dynamic attention mechanism provides them with high functional expressiveness, as they can flexibly assign different importance to neighbors based on their features. However, as we slightly increase the network depth, the attention and feature weights face difficulty changing during training, which prevents us from learning deeper and more complex models.

We have derived an explanation of this issue in the form of a structural relationship between the gradients and parameters that are associated with a feature. This relationship implies a conservation law that preserves a sum of the respective squared weight and attention norms during gradient flow. Accordingly, if weight and attention norms are highly unbalanced as is the case in standard GAT initialization schemes, relative gradients for larger parameters do not have sufficient room to increase.

This phenomenon is similar in nature to the neural tangent kernel (NTK) regime [24, 54], where only the last linear layer of a classic feed-forward neural network architecture can adapt to a task. Conservation laws for basic feed-forward architectures [15, 3, 32] do not require an infinite width assumption like NTK-related theory and highlight more nuanced issues for trainability. Furthermore, they are intrinsically linked to implicit regularization effects during gradient descent [41]. Our results are of independent interest also in this context, as we incorporate the attention mechanism into the analysis, which has implications for sequence models and transformers as well.

One of these implications is that an unbalanced initialization hampers the trainability of the involved parameters. Yet, the identification of the cause already contains an outline for its solution. Balancing the initial norms of feature and attention weights leads to more effective parameter changes and significantly faster convergence during training, even in shallow architectures. To further increase the trainability of feature weights, we endow them with a balanced orthogonal looks-linear structure, which induces perfect dynamical isometry in perceptrons [10] and thus enables signal to pass even through very deep architectures. Experiments on multiple benchmark datasets have verified the validity of our theoretical insights and isolated the effect of different modifications of the initial parameters on the trainability of GATs.

## 5   Acknowledgements

We gratefully acknowledge funding from the European Research Council (ERC) under the Horizon Europe Framework Programme (HORIZON) for proposal number 101116395 SPARSE-ML.

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

# 6 Proofs of Theorems

**Notation.** Scalar and element-wise products are denoted $\langle , \rangle$ and $\odot$, respectively. Boldface lowercase and uppercase symbols represent vectors and matrices, respectively.

Our proof of Theorem 2.2 utilizes a rescale invariance that follows from Noether's theorem, as stated by [31]. Note that we could also derive the gradient structure directly from the derivatives, but the rescale invariance is easier to follow.

**Definition 6.1** (Rescale invariance). The loss $\mathcal{L}(\theta)$ is rescale invariant with respect to disjoint subsets of the parameters $\theta_1$ and $\theta_2$ if for every $\lambda > 0$ we have $\mathcal{L}(\theta) = \mathcal{L}((\lambda\theta_1, \lambda^{-1}\theta_2, \theta_d))$, where $\theta = (\theta_1, \theta_2, \theta_d)$.

We frequently utilize the following relationship.

**Lemma 6.2** (Gradient structure due to rescale invariance [31]). *The rescale invariance of $\mathcal{L}$ enforces the following geometric constraint on the gradients of the loss with respect to its parameters:*

$$\langle \theta_1, \nabla_{\theta_1}\mathcal{L} \rangle - \langle \theta_2, \nabla_{\theta_2}\mathcal{L} \rangle = 0. \tag{10}$$

We first use this rescale invariance to prove our main theorem for a GAT with shared feature transformation weights and a single attention head, as described in Def. 2.1. The underlying principle generalizes to other GAT versions as well, as we exemplify with two other variants. Firstly, we study the case of unshared weights for feature transformation of source and target nodes. Secondly, we discuss multiple attention heads.

**Proof of Theorem 2.2** Our main proof strategy relies on identifying a multiplicative rescale invariance in GATs that allows us to apply Lemma 6.2. We identify rescale invariances for every neuron $i$ at layer $l$ that induce the stated gradient structure.

Specifically, we define the components of in-coming weights to the neuron as $\theta_1 = \{w \mid w \in W^l[i, :]\}$ and the union of all out-going edges (regarding features and attention) as $\theta_2 = \{w \mid w \in W^{l+1}[:, i]\} \cup \{a^l[i]\}$. It is left to show that these parameters are invariant under rescaling.

Let us, therefore, evaluate the GAT loss at $\lambda\theta_1$ and $\lambda^{-1}\theta_2$ and show that it remains invariant for any choice of $\lambda > 0$. Note that the only components of the network that potentially change under rescaling are $h_u^l[i]$, $h_v^{l+1}[j]$, and $\alpha_{uv}^l$. We denote the scaled network components with a tilde resulting in $\tilde{h}_u^l[i]$, $\tilde{h}_v^{l+1}[k]$, and $\tilde{\alpha}_{uv}^l$ As we show, parameters of upper layers remain unaffected, as $\tilde{h}_v^{l+1}[k]$ coincides with its original non-scaled variant $\tilde{h}_v^{l+1}[k] = h_v^{l+1}[k]$.

Let us start with the attention coefficients. Note that $\tilde{a}^l[i] = \lambda^{-1}a^l[i]$ and $\tilde{W}^l[i, j] = \lambda W^l[i, j]$. This implies that

$$\tilde{\alpha}_{uv}^l = \frac{\exp(e_{uv}^l)}{\sum_{u' \in \mathcal{N}(v)} \exp(e_{uv}^l)} = \alpha_{uv}^l \ , \quad \text{because} \tag{11}$$

$$\tilde{e}_{uv}^l = (a^l)^\top \cdot \phi_2(W^l(h_u^{l-1} + h_v^{l-1})) = e_{uv}^l, \tag{12}$$

which follows from the positive homogeneity of $\phi_2$ that allows

$$\tilde{e}_{uv}^l = \lambda^{-1}a^l[i]\phi_2(\sum_j^{n_{l-1}} \lambda W^l[i, j](h_u^{l-1}[j] + h_v^{l-1}[j])) + \sum_{i' \neq i}^{n_l} a^l[i]\phi_2(\sum_j^{n_{l-1}} W^l[i, j](h_u^{l-1}[j] + h_v^{l-1}[j])) \tag{13}$$

$$= \lambda^{-1}\lambda a^l[i]\phi_2(\sum_j^{n_{l-1}} W^l[i, j](h_u^{l-1}[j] + h_v^{l-1}[j])) + \sum_{i' \neq i}^{n_l} a^l[i]\phi_2(\sum_j^{n_{l-1}} W^l[i, j](h_u^{l-1}[j] + h_v^{l-1}[j])) \tag{14}$$

$$= e_{uv}^l. \tag{15}$$

Since $\tilde{\alpha}_{uv}^l = \alpha_{uv}^l$, it follows that

$$\tilde{h}_u^l[i] = \phi_1\left(\sum_{z \in \mathcal{N}(u)} \alpha_{zu}^l \sum_j^{n_{l-1}} \lambda W^l[i, j]h_z^{l-1}[j]\right)$$

$$= \lambda\phi_1\left(\sum_{z \in \mathcal{N}(u)} \alpha_{zu}^l \sum_j^{n_{l-1}} W^l[i, j]h_z^{l-1}[j]\right)$$

$$= \lambda h_u^l[i]$$

In the next layer, we therefore have

$$\tilde{h}_v^{l+1}[k] = \phi_1\left(\sum_{u \in \mathcal{N}(v)} \alpha_{uv}^{l+1} \sum_i^{n_l} \lambda^{-1}W^{l+1}[k, i]\tilde{h}_u^l[i]\right)$$

$$= \phi_1\left(\sum_{u \in \mathcal{N}(v)} \alpha_{uv}^{l+1} \sum_i^{n_l} \lambda^{-1}W^{l+1}[k, i]\lambda h_u^l[i]\right)$$

$$= \phi_1\left(\sum_{u \in \mathcal{N}(v)} \alpha_{uv}^{l+1} \sum_i^{n_l} W^{l+1}[k, i]h_u^l[i]\right)$$

$$= h_v^{l+1}[k]$$

Thus, the output node representations of the network remain unchanged, and according to Def. 6.1, the loss $\mathcal{L}$ is rescale-invariant.

Consequently, as per Lemma 6.2, the constraint in Eq.(10) can be written as:

$$\langle W^l[i,:], \nabla_{W^l[i,:]}\mathcal{L}\rangle - \langle a^l[i], \nabla_{a^l[i]}\mathcal{L}\rangle - \langle W^{l+1}[:,i], \nabla_{W^{l+1}[:,i]}\mathcal{L}\rangle = 0.$$

which can be rearranged to Eq.((2.2):

$$\langle W^l[i,:], \nabla_{W^l[i,:]}\mathcal{L}\rangle - \langle a^l[i], \nabla_{a^l[i]}\mathcal{L}\rangle = \langle W^{l+1}[:,i], \nabla_{W^{l+1}[:,i]}\mathcal{L}\rangle.$$

thus proving Theorem 2.2.

**Proof of Corollary 2.3**   Given that gradient flow applied on loss $\mathcal{L}$ is captured by the differential equation

$$\frac{\mathrm{d}w}{\mathrm{d}t} = -\nabla_w \mathcal{L} \tag{16}$$

which implies:

$$\frac{\mathrm{d}}{\mathrm{d}t}\left\|W^l[i,:]\right\|^2 = 2\langle W^l[i,:], \frac{\mathrm{d}}{\mathrm{d}t}W^l[i,:]\rangle = -2\langle W^l[i,:], \nabla_{W^l[i,:]}\mathcal{L}\rangle \tag{17}$$

substituting in Eq.(4) similarly for gradient flow of $\mathbf{W}^{l+1}[:,i]$ and $a^l[i]$, as done in Eq.(17) yields Theorem 2.2:

$$\frac{\mathrm{d}}{\mathrm{d}t}\left(\left\|W^l[i,:]\right\|^2 - \left\|a^l[i]\right\|^2\right) = \frac{\mathrm{d}}{\mathrm{d}t}\left(\left\|W^{l+1}[:i]\right\|^2\right).$$
$$-2\langle W^l[i,:], \nabla_{W^l[i,:]}\mathcal{L}\rangle - (-2)\langle a^l[i], \nabla_{a^l[i]}\mathcal{L}\rangle = -2\langle W^{l+1}[:,i], \nabla_{W^{l+1}[:,i]}\mathcal{L}\rangle.$$

Therefore, Eq.(4) and consequently Eq.(5) in Corollary 2.3 hold.

Summing over $i \in [n_l]$ on both sides of Eq.(4) yields Corollary 2.4.

**Theorem 6.3** (Structure of gradients for GAT without weight-sharing). *Let a GAT network as defined by Def. 2.1 consisting of $L$ layers be given. The feature transformation parameters $\mathbf{W}_s^l$ and $\mathbf{W}_t^l$ of the source and target nodes of an edge and attention parameters $\mathbf{a}^l$ of a GAT layer $l$ are defined according to Eq.(1) and (2).*

*Then the gradients for layer $l \in [L-1]$ in the network are governed by the following law:*

$$\langle W_s^l[i,:], \nabla_{W_s^l[i,:]}\mathcal{L}\rangle + \langle W_t^l[i,:], \nabla_{W_t^l[i,:]}\mathcal{L}\rangle - \langle a^l[i], \nabla_{a^l[i]}\mathcal{L}\rangle = \langle W_s^{l+1}[:,i], \nabla_{W_s^{l+1}[:,i]}\mathcal{L}\rangle \tag{18}$$

*Proof.* The proof is analogous to the derivation of Theorem 2.2. We follow the same principles and define the disjoint subsets $\theta_1$ and $\theta_2$ of the parameter set $\theta$, associated with a neuron $i$ in layer $l$ accordingly, as follows:

$$\theta_1 = \{w|w \in W_s^l[i,:]\} \cup \{w|w \in W_t^l[i,:]\}$$
$$\theta_2 = \{w|w \in W_s^{l+1}[:,i]\} \cup \{a^l[i]\}$$

Then, the invariance of node representations follows similarly to the proof of Theorem 2.2.

The only components of the network that potentially change under rescaling are $h_u^l[i]$, $h_v^{l+1}[j]$, and $\alpha_{uv}^l$. We denote the scaled network components with a tilde resulting in $\tilde{h}_u^l[i]$, $\tilde{h}_v^{l+1}[k]$, and $\tilde{\alpha}_{uv}^l$ As we show, parameters of upper layers remain unaffected, as $\tilde{h}_v^{l+1}[k]$ coincides with its original non-scaled variant $\tilde{h}_v^{l+1}[k] = h_v^{l+1}[k]$.

Let us start with the attention coefficients. Note that $\tilde{a}^l[i] = \lambda^{-1}a^l[i]$, $\tilde{W}_s^l[i,j] = \lambda W_s^l[i,j]$ and $\tilde{W}_t^l[i,j] = \lambda W_t^l[i,j]$. This implies that

$$\tilde{\alpha}_{uv}^l = \frac{\exp(e_{uv}^l)}{\sum_{u' \in \mathcal{N}(v)} \exp(e_{uv}^l)} = \alpha_{uv}^l \ , \quad \text{because} \tag{19}$$

$$\tilde{e}_{uv}^l = (a^l)^\top \cdot \phi_2(W_s^l h_u^{l-1} + W_t^l h_v^{l-1}) = e_{uv}^l, \tag{20}$$

which follows from the positive homogeneity of $\phi_2$ that allows

$$\tilde{e}^l_{uv} = \lambda^{-1} a^l[i] \phi_2 \left( \sum_j^{n_{l-1}} \lambda W_s^l[i,j] h_u^{l-1}[j] + \lambda W_t^l[i,j] h_v^{l-1}[j] \right) \tag{21}$$

$$+ \sum_{i' \neq i}^{n_l} a^l[i] \phi_2 \left( \sum_j^{n_{l-1}} W_s^l[i,j] h_u^{l-1}[j] + W_t^l[i,j] h_v^{l-1}[j] \right) \tag{22}$$

$$= \lambda^{-1} \lambda a^l[i] \phi_2 \left( \sum_j^{n_{l-1}} W_s^l[i,j] h_u^{l-1}[j] + W_t^l[i,j] h_v^{l-1}[j] \right) \tag{23}$$

$$+ \sum_{i' \neq i}^{n_l} a^l[i] \phi_2 \left( \sum_j^{n_{l-1}} W_s^l[i,j] h_u^{l-1}[j] + W_t^l[i,j] h_v^{l-1}[j] \right) \tag{24}$$

$$= e^l_{uv}. \tag{25}$$

Since $\tilde{\alpha}^l_{uv} = \alpha^l_{uv}$, it follows that

$$\tilde{h}_u^l[i] = \phi_1 \left( \sum_{z \in \mathcal{N}(u)} \alpha^l_{zu} \sum_j^{n_{l-1}} \lambda W_s^l[i,j] h_z^{l-1}[j] \right)$$

$$= \lambda \phi_1 \left( \sum_{z \in \mathcal{N}(u)} \alpha^l_{zu} \sum_j^{n_{l-1}} W_s^l[i,j] h_z^{l-1}[j] \right)$$

$$= \lambda h_u^l[i]$$

In the next layer, we therefore have

$$\tilde{h}_v^{l+1}[k] = \phi_1 \left( \sum_{u \in \mathcal{N}(v)} \alpha^{l+1}_{uv} \sum_i^{n_l} \lambda^{-1} W_s^{l+1}[k,i] \tilde{h}_u^l[i] \right)$$

$$= \phi_1 \left( \sum_{u \in \mathcal{N}(v)} \alpha^{l+1}_{uv} \sum_i^{n_l} \lambda^{-1} W_s^{l+1}[k,i] \lambda h_u^l[i] \right)$$

$$= \phi_1 \left( \sum_{u \in \mathcal{N}(v)} \alpha^{l+1}_{uv} \sum_i^{n_l} W_s^{l+1}[k,i] h_u^l[i] \right)$$

$$= h_v^{l+1}[k]$$

Thus, the application of Lemma 6.2 derives Theorem 6.3. $\qquad \square$

**Theorem 6.4** (Structure of gradients for GAT with multi-headed attention). *Given the feature transformation parameters $W_k^l$ and attention parameters $a_k^l$ of an attention head $k \in [K]$ in a GAT layer of a $L$ layer network. Then the gradients of layer $l \in [L-1]$ respect the law:*

$$\sum_k^K \langle W_k^l[i,:], \nabla_{W_k^l[i,:]} \mathcal{L} \rangle - \sum_k^K \langle a_k^l[i], \nabla_{a_k^l[i]} \mathcal{L} \rangle = \sum_k^K \langle W_k^{l+1}[:,i], \nabla_{W_k^{l+1}[:,i]} \mathcal{L} \rangle. \tag{26}$$

*Proof.* Each attention head is independently defined by Eq.1 and Eq.2, and thus Theorem 2.2 holds for each head, separately. The aggregation of multiple heads in a layer is done over node representations of each head in either one of two ways:

Concatenation: $h_v^l = \|_k^K \phi(\sum_{u \in \mathcal{N}(v)} \alpha_{kuv}{}^l \cdot W_k^l h_u^{l-1})$ , or

Average: $h_v^l = \frac{1}{K} \sum_k^K \phi(\sum_{u \in \mathcal{N}(v)} \alpha_{kuv}{}^l \cdot W_k^l h_u^{l-1})$

Thus, the rescale invariance and consequent structure of gradients of the parameters in each head are not altered and Eq.(6.2) holds by summation of the conservation law over all heads. $\qquad \square$

For the most general case of a GAT layer without weight-sharing and with multi-headed attention, each term in Theorem 6.3 is summed over all heads. Further implications analogous to Corollary 2.3, and 2.4 can be derived for these variants by similar principles.

## 7   Training Dynamics

In the main paper, we have shared observations regarding the learning dynamics of GAT networks of varying depths with standard initialization and or balanced initialization schemes. Here, we report additional figures for more varied depths to complete the picture.

All figures in this section correspond to a $L$ layer GAT network trained on Cora. For SGD, the learning rate is set to $\gamma = 0.1$, $\gamma = 0.05$ and $\gamma = 0.01$ for $L = 5$, $L = 10$ and $L = 20$, respectively. For Adam, $\gamma = 0.005$ and $\gamma = 0.0001$ for $L = 5$ and $L = 10$, respectively.

Relative change of a feature transformation parameter $w$ is defined as $|(w^* - w_0)/w^*|$ where $w_0$ is the initialized value and $w^*$ is the value for the model with maximum validation accuracy during training. Absolute change $|a^* - a_0|$ is used for attention parameters $a$ since attention parameters are initialized with 0 in the balanced initialization. We view the fraction of parameters that remain unchanged during training separately in Figure 8, and examine the layer-wise distribution of change in parameters in Figure 11 without considering the unchanged parameters.

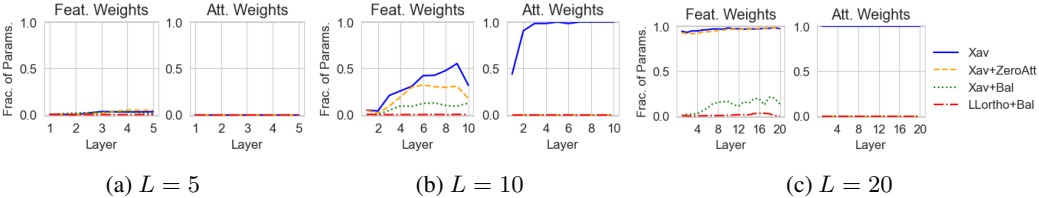

(a) $L = 5$        (b) $L = 10$        (c) $L = 20$

Figure 8: Layer-wise fraction of feature transformation parameters $W^l$ and attention parameters $a^l$ with zero relative and absolute change, respectively, trained with SGD. For $L = 2$, no parameters with zero change existed. A small number of feature weights do not change in a 5 layer unbalanced network initialized with Xavier, but this fraction becomes significantly large when depth is increased to $L = 10$. Note that $W^1$ contains a much larger number of parameters compared to the intermediate layers (specifically in this case, though it is generally common). At $L = 20$, nearly all $w \in W^l$ with an unbalanced initialization (Xav. and Xav+ZeroAtt) struggle to change during training, whereas the balanced Xavier and LL-orthogonal initialization are able to drive change in most $w \in W^l$ and all $a \in a^l$ parameters, allowing the network to train.

To analyze how gradients vary in the network during training, we define the relative gradient norm of feature transformation and attention parameters for a layer $l$ as $\left\|\nabla_{W^l} W^l\right\|_F / \left\|W^l\right\|_F$ and $\left\|\nabla_{a^l} a^l\right\| / \left\|a^l\right\|$, respectively. Figures 3b and 4b, and Figure 12 depict relative gradient norms for training under SGD and Adam respectively.

## 8   Additional Results

The results of the main paper have focused on the vanilla GAT having ReLU activation between consecutive layers, a single attention head, and shared weights for feature transformation of the source and target nodes, optimized with vanilla gradient descent (or Adam) without any regularization. Here, we present additional results for training with architectural variations, comparing a balanced initialization with a normalization scheme focused on GATs, and discussing the impact of an orthogonal initialization and the applicability of the derived conservation law to other message-passing GNNs (MPGNNs).

**Training variations**   To understand the impact of different architectural variations, common regularization strategies such as dropout and weight decay, and different activation functions, we conducted additional experiments. For a 10-layer network with width 64, Table 3 and 4 report our results for SGD and Adam, respectively. The values of hyperparameters were taken from [48].

Table 3: Mean test accuracy(%) $\pm$ 95% CI over five runs of GAT trained on Cora using SGD

| Variation | $L$ | Xav | Bal$_X$ | Bal$_O$ |
|---|---|---|---|---|
| None (Vanilla GAT) | 5 | $73.00 \pm 3.02$ | $76.96 \pm 2.21$ | $\mathbf{79.48 \pm 0.43}$ |
| | 10 | $25.48 \pm 18.13$ | $77.72 \pm 1.49$ | $\mathbf{79.46 \pm 1.34}$ |
| attention heads = 8 | 5 | $73.56 \pm 2.71$ | $77.44 \pm 1.54$ | $\mathbf{79.58 \pm 0.53}$ |
| | 10 | $25.50 \pm 18.18$ | $77.02 \pm 2.76$ | $\mathbf{79.06 \pm 0.73}$ |
| activation = ELU | 5 | $75.68 \pm 1.80$ | $79.20 \pm 1.07$ | $\mathbf{79.64 \pm 0.36}$ |
| | 10 | $73.02 \pm 2.27$ | $\mathbf{78.64 \pm 1.72}$ | $47.76 \pm 7.39$ |
| dropout = 0.6 | 5 | $42.14 \pm 15.97$ | $79.18 \pm 1.17$ | $\mathbf{81.00 \pm 0.62}$ |
| | 10 | $24.90 \pm 9.50$ | $30.94 \pm 1.04$ | $\mathbf{44.40 \pm 1.84}$ |
| weight decay = 0.0005 | 5 | $67.26 \pm 6.30$ | $77.36 \pm 1.74$ | $\mathbf{79.56 \pm 0.48}$ |
| | 10 | $18.78 \pm 11.96$ | $76.56 \pm 2.91$ | $\mathbf{79.40 \pm 1.15}$ |
| weight sharing = False | 5 | $70.04 \pm 3.45$ | $77.46 \pm 1.19$ | $\mathbf{79.66 \pm 0.72}$ |
| | 10 | $19.54 \pm 15.64$ | $76.16 \pm 2.28$ | $\mathbf{80.06 \pm 1.12}$ |

Table 4: Mean test accuracy(%) $\pm$ 95% CI over five runs of GAT trained on Cora using Adam

| Variation | $L$ | Xav | Bal$_X$ | Bal$_O$ |
|---|---|---|---|---|
| None (Vanilla GAT) | 5 | $76.18 \pm 1.61$ | $79.38 \pm 2.24$ | $\mathbf{80.20 \pm 0.57}$ |
| | 10 | $70.86 \pm 3.99$ | $75.72 \pm 2.35$ | $\mathbf{79.62 \pm 1.27}$ |
| attention heads = 8 | 5 | $75.62 \pm 1.74$ | $78.54 \pm 1.06$ | $\mathbf{79.56 \pm 0.85}$ |
| | 10 | $70.94 \pm 2.76$ | $75.48 \pm 2.48$ | $\mathbf{79.74 \pm 1.10}$ |
| activation = ELU | 5 | $76.56 \pm 1.72$ | $78.72 \pm 1.02$ | $\mathbf{79.56 \pm 0.64}$ |
| | 10 | $75.30 \pm 1.42$ | $\mathbf{78.48 \pm 1.79}$ | $76.52 \pm 0.97$ |
| dropout = 0.6 | 5 | $76.74 \pm 1.44$ | $78.42 \pm 1.28$ | $\mathbf{79.92 \pm 0.48}$ |
| | 10 | $32.48 \pm 6.99$ | $31.76 \pm 1.33$ | $\mathbf{76.34 \pm 0.95}$ |
| weight decay = 0.0005 | 5 | $75.10 \pm 2.05$ | $78.52 \pm 1.41$ | $\mathbf{79.80 \pm 0.63}$ |
| | 10 | $28.74 \pm 12.04$ | $74.68 \pm 3.06$ | $\mathbf{79.70 \pm 1.14}$ |
| weight sharing = False | 5 | $76.90 \pm 0.92$ | $78.90 \pm 1.07$ | $\mathbf{80.62 \pm 1.36}$ |
| | 10 | $74.46 \pm 1.90$ | $75.50 \pm 3.49$ | $\mathbf{79.78 \pm 0.70}$ |

We find that the balanced initialization performs similar in most variations, outperforming the unbalanced standard initialization by a substantial margin. Dropout generally does not seem to be helpful to the deeper network ($L = 10$), regardless of the initialization. From Table 3 and 4, it is evident that although techniques like dropout and weight decay may aid optimization, they are alone not enough to enable the trainability of deeper network GATs and thus are complementary to balancedness.

Note that ELU is not a positively homogeneous activation function, which is an assumption in out theory. In practice, however, it does not impact the Xavier-balanced initialization (Bal$_X$). However, the Looks-Linear orthogonal structure is specific to ReLUs. Therefore, the orthogonality and balancedness of the Bal$_O$ initialization are negatively impacted by ELU, although the Adam optimizer seems to compensate for it to some extent.

In addition to increasing the trainability of deeper networks, the balanced initialization matches the state-of-the-art performance of Xavier initialized 2 layer GAT, given the architecture and optimization hyperparameters as reported in [48]. We used an existing GAT implementation and training script[2] of

---

[2]https://github.com/Diego999/pyGAT

Cora that follows the original GAT(v1)[48] paper and added our balanced initialization to the code. As evident from Table 5, the balanced initialization matches SOTA performance of GAT(v1) on Cora (83-84%) (on a version of the dataset with duplicate edges). GAT(v2) achieved 78.5% accuracy on Pubmed but the corresponding hyperparameter values were not reported. Hence, we transferred them from [48]. This way, we are able to match the state-of-the-art performance of GAT(v2) on Pubmed, as shown in Table 5.

Table 5: GAT performance on Cora and Pubmed: mean test accuracy$(\%) \pm 95\%$ CI over five runs.

| | Xav | Bal$_X$ | Bal$_O$ |
|---|---|---|---|
| Cora | $84.50 \pm 0.52$ | $\mathbf{84.58 \pm 0.65}$ | $84.55 \pm 0.47$ |
| Pubmed | $78.38 \pm 0.77$ | $78.52 \pm 0.54$ | $\mathbf{78.56 \pm 0.20}$ |

**Comparison with Lipshitz Normalization**    A feasible comparison can be carried out with [14] that proposes a Lipschitz normalization technique aimed at improving the performance of deeper GATs in particular. We use their provided code to reproduce their experiments on Cora, Citeseer, and Pubmed for 2,5,10,20 and 40-layer GATs with Lipschitz normalization and compare them with LLortho+Bal initialization in Table 6. Note that Lipschitz normalization has been shown to outperform other previous normalization techniques for GNNs such as pair-norm [64] and layer-norm [5].

Table 6: Comparing a balanced LL-orthogonal initialization to Lipschitz normalization applied with a standard (imbalanced) Xavier initialization: the balanced initialization results in a much higher accuracy as the depth of the network increases.

| Dataset | Cora | | Citeseer | | Pubmed | |
|---|---|---|---|---|---|---|
| Layers | Lip. Norm. | Bal$_O$ Init. | Lip. Norm. | Bal$_O$ Init. | Lip. Norm. | Bal$_O$ Init. |
| 2 | **82.1** | 79.5 | 65.4 | **67.7** | 74.8 | **76.0** |
| 5 | 77.1 | **80.2** | 63.0 | **67.7** | 73.7 | **75.7** |
| 10 | 78.0 | **79.6** | 43.6 | **67.4** | 52.8 | **76.9** |
| 20 | 72.2 | **77.3** | 18.2 | **66.3** | 23.3 | **77.3** |
| 40 | 12.9 | **75.9** | 18.1 | **63.2** | 36.6 | **77.5** |

**Impact of orthogonal initialization**    In the main paper, we have advocated and presented results for using a balanced LL-orthogonal initialization. Here, we discuss two special cases of orthogonal initialization (and their balanced versions): identity matrices that have also been used for GNNs in [18, 16], and matrices with looks-linear structure using an identity submatrix (LLidentity) since a looks-linear structure would be more effective for ReLUs [10].

In line with [18], identity and LLidentity are used to initialize the hidden layers while Xavier is used to initialize the first and last layers. A network initialized with identity is balanced by adjusting the weights of the first and last layer to have norm 1 (as identity matrices have row-wise and column-wise norm). A network with LLidentity initialization is balanced to have norm 2 in the first and last layer, similar to LLortho initialization. We compare the performance using these four base initializations (one standard Xavier, and three orthogonal cases) and their balanced counterparts in Fig. 9.

We observe that balancing an (LL-)orthogonal initialization results in an improvement in the generalization ability of the network in most cases and speeds up training, particularly for deeper networks. However, note that an (LL-)orthogonal initialization itself also has a positive effect on trainability in particular of deep networks. Contributing to this fact is the mostly-balanced nature of an (LL-)orthogonal initialization i.e. given hidden layers of equal dimensions the network is balanced in all layers except the first and last layers (assuming zero attention parameters), which allows the model to train, as opposed to the more severely imbalanced standard Xavier initialization. This further enforces the key takeaway of our work that norm imbalance at initialization hampers the trainability of GATs. In addition, the LLortho+Bal initialization also speeds up training over the LLortho initialization even in cases in which the generalization performance of the model is at par for both initializations.

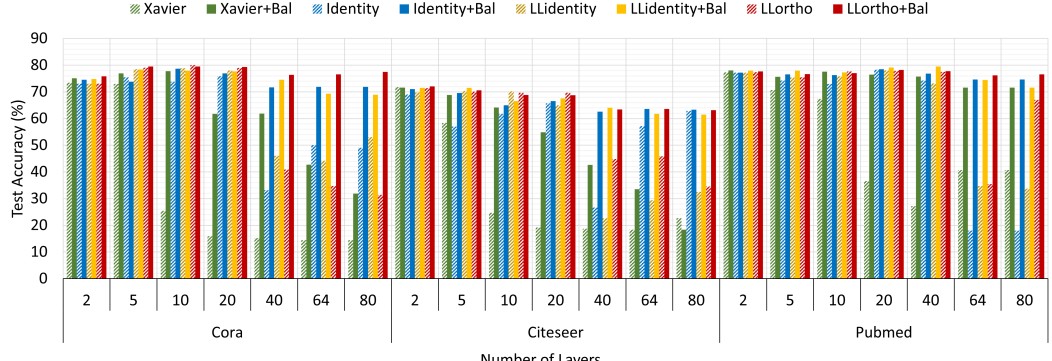

Figure 9: GAT network with varying initialization trained using SGD.

Note that identity initializations have also been explored in the context of standard feed-forward neural networks. While they tend to work in practice, the lack of induced feature diversity can be problematic from a theoretical point of view (see e.g. for a counter-example [7]). However, potentially due to the reasons regarding feature diversity and ReLU activation discussed above, the balanced looks-linear random orthogonal initialization (LLortho+Bal.) outperforms initialization with identity matrices (see Fig. 9). In most cases, the balanced versions outperform the imbalanced version of the base initialization and the performance gap becomes exceedingly large as network depth increases.

**Applicability to other MPGNNs** As GAT is a generalization of the GCN, all theorems are also applicable to GCNs (where the attention parameters are simply 0). We provide additional experiments in Table 7, comparing the performance of GCN models of varying depths with imbalanced and balanced initializations trained on Cora using the SGD optimizer. We use the same hyperparameters as reported for GAT. As expected, the trainability benefits of a balanced initialization are also evident in deeper GCN networks.

Table 7: GCN with varying initialization trained using SGD.

| Dataset | Depth | Xavier | Xavier+Bal | LLortho | LLortho+Bal |
|---------|-------|--------|------------|---------|-------------|
| Cora | 2 | $77.8 \pm 0.9$ | $80.5 \pm 0.5$ | $78.0 \pm 0.3$ | $\mathbf{80.9 \pm 0.4}$ |
| | 5 | $73.2 \pm 3.4$ | $78.3 \pm 0.8$ | $\mathbf{80.3 \pm 0.8}$ | $79.6 \pm 1.0$ |
| | 10 | $24.1 \pm 4.5$ | $77.6 \pm 2.0$ | $80.0 \pm 1.1$ | $\mathbf{80.0 \pm 0.9}$ |
| | 20 | $14.4 \pm 11.2$ | $62.8 \pm 3.6$ | $78.7 \pm 0.5$ | $\mathbf{78.8 \pm 1.5}$ |
| | 40 | $13.4 \pm 0.9$ | $65.9 \pm 7.3$ | $28.3 \pm 16.2$ | $\mathbf{77.1 \pm 0.9}$ |
| | 64 | $9.8 \pm 5.4$ | $33.0 \pm 13.4$ | $27.3 \pm 12.7$ | $\mathbf{76.7 \pm 1.3}$ |
| | 80 | $12.4 \pm 19.3$ | $33.8 \pm 12.9$ | $38.9 \pm 21.3$ | $\mathbf{77.1 \pm 1.3}$ |
| Citeseer | 2 | $66.6 \pm 20.0$ | $71.3 \pm 1.8$ | $66.0 \pm 3.2$ | $\mathbf{72.3 \pm 0.9}$ |
| | 5 | $60.9 \pm 12.3$ | $66.9 \pm 15.0$ | $69.0 \pm 6.4$ | $\mathbf{70.1 \pm 1.8}$ |
| | 10 | $23.8 \pm 36.8$ | $66.0 \pm 5.9$ | $\mathbf{70.6 \pm 0.9}$ | $69.8 \pm 10.9$ |
| | 20 | $16.4 \pm 18.2$ | $47.9 \pm 10.0$ | $67.0 \pm 8.6$ | $\mathbf{69.7 \pm 4.5}$ |
| | 40 | $13.9 \pm 56.8$ | $37.3 \pm 92.8$ | $44.8 \pm 6.8$ | $\mathbf{64.7 \pm 13.6}$ |
| | 64 | $13.8 \pm 41.4$ | $29.5 \pm 15.0$ | $37.3 \pm 79.6$ | $\mathbf{66.3 \pm 0.5}$ |
| | 80 | $12.4 \pm 42.7$ | $25.8 \pm 3.6$ | $30.1 \pm 21.8$ | $\mathbf{64.1 \pm 3.2}$ |

The underlying principle of a balanced network initialization holds in general. However, adapting the balanced initialization to different methods entails modification of the conservation law derived for GATs to correspond to the specific architecture of the other method. For example, the proposed balanced initialization can be applied directly to a more recent variant of GAT, $\omega$GAT [18], for which the same conservation law holds and also benefits from a balanced initialization. We conduct

additional experiments to verify this. The results in Fig. 10 follow a similar pattern as for GATs: a balanced orthogonal initialization with looks linear structure (LLortho+Bal) of $\omega$GAT performs the best, particularly by a wide margin at much higher depths (64 and 80 layers).

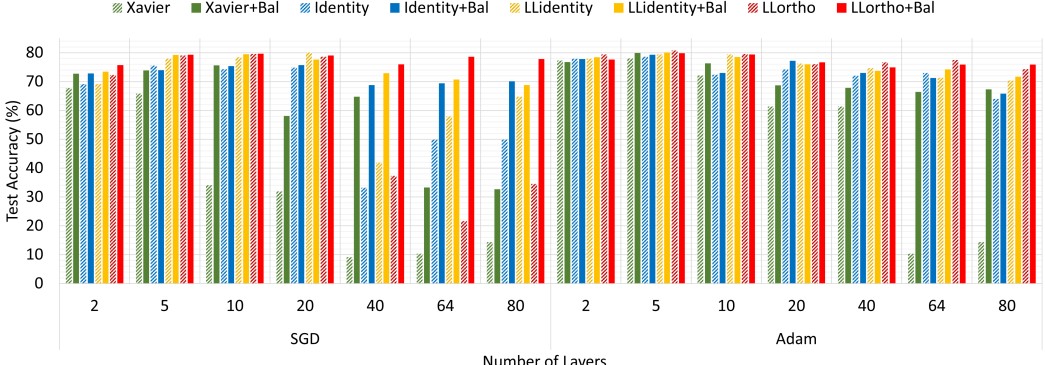

Figure 10: $\omega$GAT network with varying trained on Cora.

In this work, we focus our exposition on GATs and take the first step in modeling the training dynamics of attention-based models for graph learning. An intriguing direction for future work is to derive modifications in the conservation law for other attention-based models utilizing the dot-product self-attention mechanism such as SuperGAT [28] and other architectures based on Transformers[47], that are widely used in Large Language Models (LLMs) and have recently also been adapted for graph learning [58, 60].

## 9 Datasets Summary

We used nine benchmark datasets for the transductive node classification task in our experiments, as summarized in Table 8.

For the Platenoid datasets (Cora, Citeseer, and Pubmed) [56], we use the graphs provided by Pytorch Geometric (PyG), in which each link is saved as an undirected edge in both directions. However, the number of edges is not exactly double (for example, $5429$ edges are reported in the raw Cora dataset, but the PyG dataset has $10556 < (2 \times 5429))$ edges as duplicate edges have been removed in preprocessing. We also remove isolated nodes in the Citeseer dataset.

The WebKB (Cornell, Texas, and Wisconsin), Wikipedia (Squirrel and Chameleon) and Actor datasets [39], are used from the replication package provided by [40], where duplicate edges are removed.

All results in this paper are according to the dataset statistics reported in Table 8.

Table 8: Summary of datasets used in experiments.

| Dataset | Cora | Cite. | Pubmed | Cham. | Squi. | Actor | Corn. | Texas | Wisc. |
|---|---|---|---|---|---|---|---|---|---|
| # Nodes | 2708 | 3327 | 19717 | 2277 | 5201 | 7600 | 183 | 183 | 251 |
| # Edges | 10556 | 9104 | 88648 | 31371 | 198353 | 26659 | 277 | 279 | 450 |
| # Features | 1433 | 3703 | 500 | 2325 | 2089 | 932 | 1703 | 1703 | 1703 |
| # Classes | 7 | 6 | 3 | 5 | 5 | 5 | 5 | 5 | 5 |
| # Train | 140 | 120 | 60 | 1092 | 2496 | 3648 | 87 | 87 | 120 |
| # Validation | 500 | 500 | 500 | 729 | 1664 | 2432 | 59 | 59 | 80 |
| # Test | 1000 | 1000 | 1000 | 456 | 1041 | 1520 | 37 | 37 | 51 |

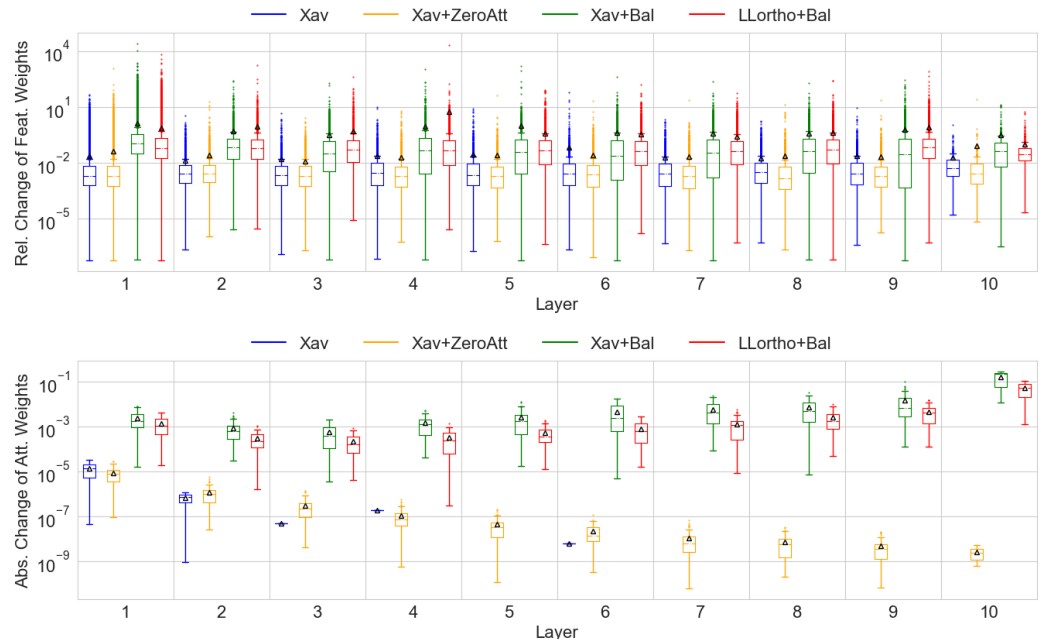

(a) $L = 10$: The balanced initialization allows larger changes in a higher number of parameters in $W^l$ across all layers $l$ with the highest margin in $l = 1$. The change distribution for the parameters in $a^l$ is missing for $l = 5$ and $l \in [7, 10]$ because these parameters remain unchanged (see Fig.8).

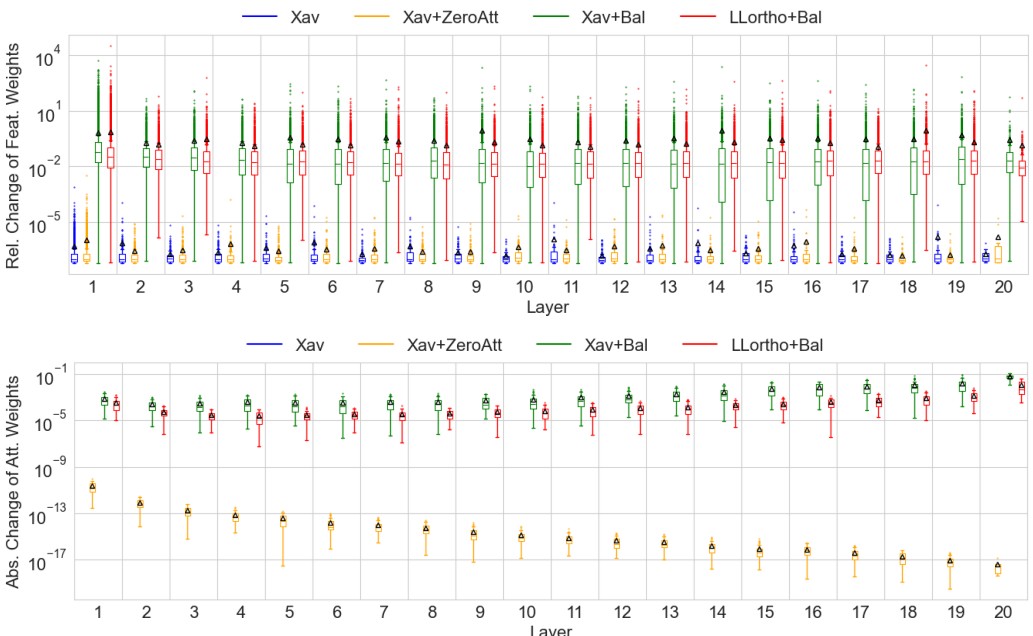

(b) $L = 20$: The same pattern as in $L = 10$ is seen. However, the difference between the trainability of models with unbalanced and balanced initialization becomes even more prominent. No attention parameters change at all with standard Xavier initialization (and hence their change distribution is not visible in the plot).

Figure 11: Distributions of the relative and absolute change in the values of feature transformation parameters $W^l$ and attention parameters $a^l$, respectively, when trained using SGD with unbalanced (Xav. and Xav+ZeroAtt) and balanced (Xav+Bal and LLortho+Bal) initialization. The black markers in each standard box-whisker plot denote the mean. In general, larger changes occur in attention parameters at later layers closer to the output of the network, whereas feature parameters change more in the earlier layers at the input of the network. We observe this also from the perspective of relative gradients, as higher gradients cause higher parameter changes.

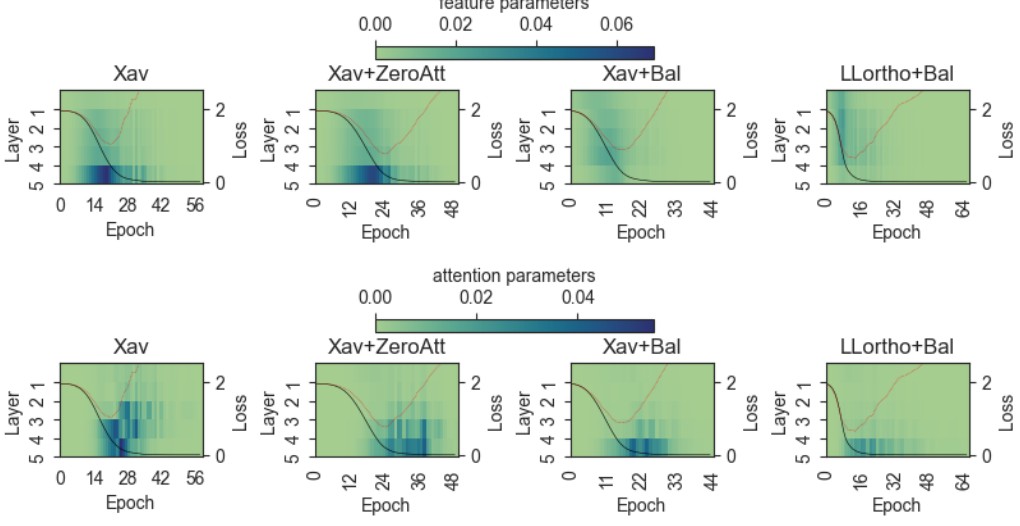

(a) $L = 5$: test accuracy (left to right) is 71.6%, 76.2%, 77.3%, and 80.5%.

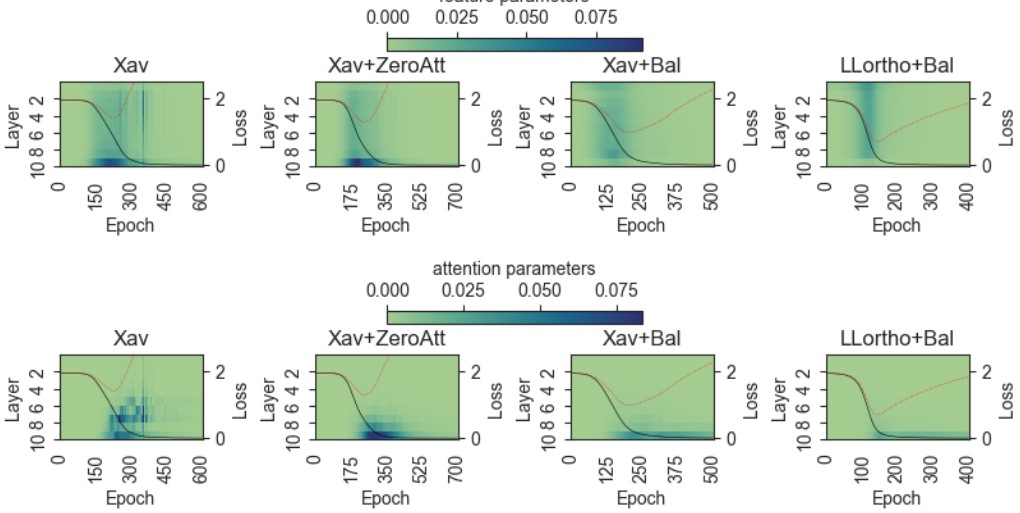

(b) $L = 10$: test accuracy (left to right) is 63.2%, 68.8%, 71.8%, and 78.0%.

Figure 12: Layer-wise relative gradient norms of GAT parameters trained with Adam, which in itself improves trainability over SGD, especially for $L = 10$ (see Fig.4b) even with standard initialization. However, with a balanced initialization, the training is faster and the model also generalizes better, as evident from the significant margin in test accuracy. A general noticeable trend, particularly with balanced initialization, is that the attention parameters have higher gradients at the end of the network (and thus change more in the later layers (see Fig. 11)) whereas the feature parameters have high gradients at the start of the network. Also, it can be seen that the feature parameters begin to change a few epochs before the attention parameters change noticeably, while the attention parameters continue to change for a few more epochs after the gradients of feature parameters have dropped. We speculate a representational benefit could drive this behaviour, i.e. with depth, the learned representation of neighbors becomes increasingly informative for a node and thus leads to a higher activity of the attention mechanism.

