# OpenReview forum: "Are GATs Out of Balance?"
_NeurIPS.cc/2023/Conference — NeurIPS 2023 poster_

### Official Review · Reviewer_hDWR · 2023-07-02

**Soundness:** 3 good
**Presentation:** 3 good
**Contribution:** 3 good
**Rating:** 6
**Confidence:** 3

**Summary:**

The work derives a conservation for the dynamics of the GAT gradient flow during training. The conservation law is used to explain why it is challenging to train deep GAT models and in particular why a large portion of parameters do not change much throughout training. A new initialization scheme is introduced to mitigate these issues.

**Strengths:**

Training GATs is notoriously problematic and this is especially the case with many layers. The work takes an interesting angle of trying to further explain why this is the case by studying conservation laws induced by the gradient flow during training. The observation that weight gradients must be small in order to satisfy such a conservation law is very interesting and practically useful. The balancing algorithm proposed is also practically useful and seems to perform well especially with deep GATs.

**Weaknesses:**

While the work focuses on GAT, it would be interesting to have a statement on more general classes of MPNNs. There are some faint connections in the paper to GCN for instance, but it would be interesting to have a more general result as well. Experimentally this might also be interesting to see if such an initialisation can be adapted to different types of MPNNs.

**Questions:**

Would the proposed initialisation technique work well for other types of MPNNs (with appropriate modifications)? Would you have experiments/theory to back this up?

**Limitations:**

The authors address the fact that the derived theory defines a conservation law that does not explain detailed dynamics but still remains relatively coarse. As such it is not able to explain some phenomena as why attention parameters change most on the first layer. I still believe that the work is a good step in helping explain phenomena in this direction.

---

> ### Author Rebuttal · Authors · 2023-08-09
>
> We thank reviewer hDWR for their interesting comments on our work and the constructive feedback.
> In response to their question, we discuss the adaptability of the balanced initialization scheme to other MPGNNs, which we would be happy to add to the manuscript.
>
> While in principle the argument of imbalanced-ness is applicable to nearly all types of MPGNNs, initializing the model in a balanced manner requires deriving conservation laws inherent to the specific architecture.
> Nevertheless, we outline a number of cases in which our derived conservation law (and consequent balanced initialization) is directly applicable.
>
> Firstly, we note that GAT is a generalization of the GCN, and therefore all theorems are also applicable to GCNs (where the attention parameters are simply 0).
> We provide additional experiments in the following table, comparing the performance of GCN models of varying depths with imbalanced and balanced initializations trained on Cora using the SGD optimizer. We use the same hyperparameters that we reported in the paper.
>
> Table 1: **GCN** trained on **Cora** using SGD:
>
> | Depth  | Xavier                 | Xavier+Bal            | LLortho               | LLortho+Bal           |
> | :----: | :--------------------: | :-------------------: | :-------------------: | :-------------------: |
> | 2  | 77\.8 &pm; 0.9   | 80\.5 &pm; 0.5  | 78\.0 &pm; 0.3  | **80\.9 &pm; 0.4**  |
> | 5  | 73\.2 &pm; 3.4   | 78\.3 &pm; 0.8  | **80\.3 &pm; 0.8**  | 79\.6 &pm; 1.0  |
> | 10 | 24\.1 &pm; 4.5   | 77\.6 &pm; 2.0  | 80\.0 &pm; 1.1  | **80\.0 &pm; 0.9**  |
> | 20 | 14\.4 &pm; 11.2  | 62\.8 &pm; 3.6  | 78\.7 &pm; 0.5  | **78\.8 &pm; 1.5** |
> | 40 | 13\.4 &pm; 0.9   | 65\.9 &pm; 7.3  | 28\.3 &pm; 16.2 | **77\.1 &pm; 0.9**  |
> | 64 | 9\.8 &pm; 5.4    | 33\.0 &pm; 13.4 | 27\.3 &pm; 12.7 | **76\.7 &pm; 1.3**  |
> | 80 | 12\.4 &pm; 19.3  | 33\.8 &pm; 12.9 | 38\.9 &pm; 21.3 | **77\.1 &pm; 1.3**  |
>
> Table 2: **GCN** trained on **Citeseer** using SGD:
>
> | Depth  | Xavier                 | Xavier+Bal            | LLortho               | LLortho+Bal           |
> | :----: | :--------------------: | :-------------------: | :-------------------: | :-------------------: |
> | 2  | 66\.6 &pm; 20.0  | 71\.3 &pm; 1.8  | 66\.0 &pm; 3.2  | **72\.3 &pm; 0.9**  |
> | 5  | 60\.9 &pm; 12.3  | 66\.9 &pm; 15.0 | 69\.0 &pm; 6.4  | **70\.1 &pm; 1.8**  |
> | 10 | 23\.8 &pm; 36.8  | 66\.0 &pm; 5.9  | **70\.6 &pm; 0.9**  | 69\.8 &pm; 10.9 |
> | 20 | 16\.4 &pm; 18.2  | 47\.9 &pm; 10.0 | 67\.0 &pm; 8.6  | **69\.7 &pm; 4.5**  |
> | 40 | 13\.9 &pm; 56.8  | 37\.3 &pm; 92.8 | 44\.8 &pm; 6.8  | **64\.7 &pm; 13.6** |
> | 64 | 13\.8 &pm; 41.4  | 29\.5 &pm; 15.0 | 37\.3 &pm; 79.6 | **66\.3 &pm; 0.5**  |
> | 80 |  12\.4 &pm; 42.7 | 25\.8 &pm; 3.6  | 30\.1 &pm; 21.8 | **64\.1 &pm; 3.2**  |
>
> Secondly, we recall that we cater to several architectural variations within the GAT architecture itself. For example, both the GATv1 [1] and GATv2 [2] have the same conservation law and we derive more general versions of the law for two GAT variants:
> i) unshared weights for feature transformation of source and target nodes, and
> ii) multiple attention heads. Furthermore, residual skip connections between layers are also supported in a balanced initialization provided their parameters are initialized with zero.
>
> Lastly, our derived conservation law also holds for a more recent architectural variation of GAT, the **$\omega$GAT** [3], which also seems to benefit from a balanced initialization.
> We conduct additional experiments to verify this. The results follow a similar pattern as for GATs (See **Figure 1(b)** of the attached PDF).
> A balanced orthogonal initialization with looks linear structure (LLortho+Bal) of $\omega$GAT performs the best, particularly by a wide margin at much higher depths (64 and 80 layers).
>
> In this work, we focus our exposition on GATs and take the first step in modeling the training dynamics of attention-based models for graph learning. An intriguing direction for future work is to derive modifications in the conservation law for other attention-based models such as SuperGAT [4] and Transformers which both utilize the dot-product self-attention mechanism.
>
> [1] Petar Veličković et al. Graph Attention Networks. In International Conference on Learning Representations, 2018.
>
> [2] Shaked Brody, Uri Alon, and Eran Yahav. How attentive are graph attention networks? In International Conference on Learning Representations, 2022.
>
> [3] Dongkwan Kim and Alice Oh. How to find your friendly neighborhood: Graph attention design with self-supervision. In International Conference on Learning Representations, 2021.
>
> [4] Moshe Eliasof, Lars Ruthotto, and Eran Treister. Improving Graph Neural Networks with Learnable Propagation Operators. In International Conference on Machine Learning, 2023.

---

> > ### Comment · Reviewer_hDWR · 2023-08-11
> >
> > Thank you for the response and the additional experiments with GCN. The smoothing results with GCN indeed seem quite promising. This is an interesting approach to a useful problem and I recommend accepting the work.

---

### Official Review · Reviewer_d9X8 · 2023-07-04

**Soundness:** 4 excellent
**Presentation:** 3 good
**Contribution:** 3 good
**Rating:** 6
**Confidence:** 3

**Summary:**

This paper focuses on the parameter struggle training problem of the well-known GNN structure GAT. The authors propose to alleviate the issue via parameter norm balancedness. A conservation law is derived for GATs with positive homogeneous activation function as the theoretical support for the parameter norm balancedness based initialization method. The authors also conduct extensive experiments to prove its effectiveness and fast convergence property.

**Strengths:**

- The problem is well identified.
- The proposed theories are solid, and well serves the problem.
- The proposed initialization method is simple yet effective in improving accuracy and speeding convergence.

**Weaknesses:**

- Some explanation skips details and may raise confusions. For example, the equations in line 140-142.
- Some formats could be improved for better illustration, such as larger figure size and fitted table style.


**Questions:**

GAT mainly exploits the attention mechanism. Can this initialization be adapted to any other methods that utilize the attention mechanism?

**Limitations:**

The author may address the limitation from the generalization of the proposed method beyond the GAT structure.

---

> ### Author Rebuttal · Authors · 2023-08-09
>
>
> We thank reviewer d9X8 for their constructive feedback. In line with their suggestions, we will i) include the details of equations used in lines 140-142 arising from the definition of Xavier initialization, and ii) increase figure sizes and improve the table style for better illustration.
>
> The underlying principle of a balanced network initialization holds in general. However, adapting the balanced initialization to different methods entails modification of the conservation law derived for GATs to correspond to the specific architecture of the other method.
> For example, the proposed balanced initialization can be applied directly to a more recent variant of GAT, **$\omega$GAT** [1], for which the same conservation law holds. We conduct additional experiments on $\omega$GAT to verify this. The results follow a similar pattern as for GATs. (See **Figure 1(b)** of the attached PDF). A balanced orthogonal initialization with looks linear structure (LLortho+Bal) of $\omega$GAT performs the best, particularly by a wide margin at much higher depths (64 and 80 layers).
>
> However, the derived conservation law and consequent balanced initialization can not be directly applied to SuperGAT [2], which employs a different kind of self-attention, namely the dot-product attention similar to the Transformer architecture.
> An intriguing direction for future work is to derive modifications in the conservation law for such other attention-based models. Currently, as suggested by the reviewer, we will mention this limitation in the main paper.
>
> [1] Dongkwan Kim and Alice Oh. How to find your friendly neighborhood: Graph attention design with self-supervision. In International Conference on Learning Representations, 2021.
>
> [2] Moshe Eliasof, Lars Ruthotto, and Eran Treister. Improving Graph Neural Networks with Learnable Propagation Operators. In International Conference on Machine Learning, 2023.

---

### Official Review · Reviewer_yMby · 2023-07-05

**Soundness:** 3 good
**Presentation:** 2 fair
**Contribution:** 3 good
**Rating:** 7
**Confidence:** 4

**Summary:**

The authors propose a theoretical analysis of the initialization of GATs and their impact on the performance of such networks, focusing on the performance vs depth aspect.

The authors propose an initialization algorithm, that starts from a random initialization and modifies the initial random weights to adhere to the findings of the theoretical analysis.

The authors then show the impact of their proposed initialization on several node classification datasets. Specifically, they show that by initializing GATs with the proposed algorithm, deep GATs can be trained to achieve better than standard (Xavier) initialized GATs.


**Strengths:**

The paper addresses a real issue with GATs - the degradation of performance when more layers are added.

The theoretical analysis seems correct to me and is supported with experimental results as well as the inspection of actual training artifacts and details (e.g., the gradients of the GAT layers).



**Weaknesses:**

The paper can be slightly better written, in terms of organization. I think that adding more paragraphs/subsections to better separate between the parts of the paper can help to ease the reader.

The paper lacks a few relevant citations that also consider graph neural networks as gradient flow (see [1][2][3]). However, their focus is not on the initialization of GATs, and therefore the paper here is novel on its own.

While the experimental results are compelling and show the benefit of the proposed method, I think that the authors should also include comparisons with other methods.

[1] GRAND: Graph Neural Diffusion

[2] Understanding convolution on graphs via energies

[3] Improving Graph Neural Networks with Learnable Propagation Operators

**Questions:**

1. Can the authors add results with more layers? For instance 64,80 layers? does the method still work with very deep network?

2. Regarding the initialization of graph neural networks, it is discussed in [3][4] that the GNN weights are initialized with identity matrices. Can the authors comment on this point? is it related to the proposed method?

3. Can the proposed initialization scheme be applied to other graph attention layers, such as superGAT[5] ?

[4] PDE-GCN: Novel Architectures for Graph Neural Networks Motivated by Partial Differential Equations

[5] How to Find Your Friendly Neighborhood: Graph Attention Design with Self-Supervision


**Limitations:**

Yes

---

> ### Author Rebuttal · Authors · 2023-08-09
>
> We thank reviewer yMby for recognizing the novelty and relevance of our work and appreciate their suggestions, for which we will be taking the following actions:
> i) introduce more subsections and break down larger paragraphs to improve the readability and comprehension of the paper;
> ii) cite the proposed relevant literature [1-5];
> iii) add experiments in line with the suggestions.
>
> **Comparison with other methods:**
> We would like to highlight that the main contribution of our work is the derivation of a conservation law for GATs and the insight into how this law can explain trainability issues for standard initialization schemes.
> In line with this reasoning, it is not feasible to directly compare with [1,2,4], as they rely on different GNN architectures with potentially different inherent conservation laws.
> Since [3] proposes an architectural variant of GAT, the $\omega$GAT, our proposed balanced initialization would apply to this setting and could potentially improve the trainability of deep $\omega$GAT. Since the code for this very recent work has not been shared, we cannot provide extensive experiments in this short time frame.
> However, we do implement $\omega$GAT in our setup and the results follow a similar pattern to GAT (See **Figure 1(b)** of the attached PDF). A balanced orthogonal initialization with looks linear structure (LLortho+Bal) of $\omega$GAT performs the best, particularly by a wide margin at much higher depths (64 and 80 layers).
>
> Yet, a feasible comparison can be carried out with [6] which proposes a Lipschitz normalization technique aimed to improve the performance of deeper GATs in particular.
> We use the code provided by [6] to reproduce their experiments on Cora, Citeseer, and Pubmed for 2,5,10,20 and 40-layer GATs with Lipschitz normalization and compare them with our results of LLortho+Bal initialization (Bal$_O$ Init.), reported in the main paper, as follows:
>
> | Layers | Cora          |                  | Citeseer   |                  | Pubmed     |                  |
> | :-----: | :-----------: | :--------------: | :--------: | :--------------: | :--------: | :--------------: |
> |          | Lip. Norm.    | Bal$_O$ Init. | Lip. Norm. | Bal$_O$ Init. | Lip. Norm. | Bal$_O$ Init. |
> | 2   | **82\.1** | 79\.5        | 65\.4  | **67\.7**    | 74\.8  | **76\.0**    |
> | 5   | 77\.1     | **80\.2**    | 63     | **67\.7**    | 73\.7  | **75\.7**    |
> | 10  | 78\.0     | **79\.6**    | 43\.6  | **67\.4**    | 52\.8  | **76\.9**    |
> | 20  | 72\.2     | **77\.3**    | 18\.2  | **66\.3**    | 23\.3  | **77\.3**    |
> | 40  | 12\.9     | **75\.9**    | 18\.1  | **63\.2**    | 36\.6  | **77\.5**    |
>
> As evident from the above table, the balanced initialization results in a much higher accuracy as the depth of the network increases than the application of the Lipschitz normalization to a standard-initialized network.
> Note that Lipschitz normalization has been shown to outperform other previous normalization techniques for GNNs such as pair-norm and layer-norm.
>
> **Higher depth:**
> Additional results for 64 and 80-layer networks are presented in **Figure 1** in the additionally submitted PDF. The improved performance of models with balanced initialization as opposed to models with standard initialization is upheld even more so for very deep networks.
>
> **Initialization with identity matrix:**
> Regarding the initialization of GNNs with identity matrices, the identity matrix can be considered a special case of an orthogonal initialization.
> Given that the hidden layers are all of the same dimensions, a network initialized with identity weight matrices for the intermediate layers and zero attention parameters would be balanced for all the hidden layers (but not with respect to the first and last layers).
> However, for ReLUs a looks-linear structure where the submatrix is initialized as identity matrix would be more effective.
> Note that identity initializations have also been explored in the context of standard feed forward neural networks.
> While they tend to work in practice, the lack of induced feature diversity can be problematic from a theoretical point of view (see e.g. for a counter example [7]).
> We conduct experiments using identity matrices to initialize the hidden layers and Xavier initialization for the first (input) and last (output) layers.
> We compare this with a balanced version by adjusting the weights of the first and last layer to have norm $1$ (as identity matrices have row-wise and column-wise norm $1$).
>
> However, potentially due to the reasons regarding feature diversity and ReLU activation discussed above, the balanced looks-linear random orthogonal initialization (LLortho+Bal.) outperforms initialization with identity matrices (See **Figure 1** in additional PDF). In most cases, the balanced versions outperform the imbalanced version of the base initialization.
>
> **SuperGATs**:
> The proposed initialization cannot be directly applied to the self-attention layer used in the SuperGAT architecture [5].
> SuperGAT combines the attention layer of the original GAT architecture with the dot-product self-attention similar to the Transformer architecture.
> This requires a modification of the derived conservation law, which would be an intriguing direction for future investigations.
>
> [6] Dasoulas et al. Lipschitz Normalization for Self-Attention Layers with Application to Graph Neural Networks. In International Conference on Machine Learning, 2021.
>
> [7] Bartlett et al. Gradient descent with identity initialization efficiently learns positive definite linear transformations by deep residual networks. In International Conference on Machine Learning, 2018.
>
> [8] Chen et al. Measuring and relieving the over-smoothing problem for
> graph neural networks from the topological view. In Proceedings of the AAAI Conference on Artificial Intelligence, 2020.

---

> > ### Comment · Reviewer_yMby · 2023-08-16
> > **Thank you for the detailed rebuttal**
> >
> > I would like to thank the authors for the detailed rebuttal. Your answers addressed all my questions and therefore I am happy to increase my score.

---

### Official Review · Reviewer_cXS5 · 2023-07-07

**Soundness:** 3 good
**Presentation:** 3 good
**Contribution:** 3 good
**Rating:** 7
**Confidence:** 3

**Summary:**

This work proves a conservation law for GAT architectures, which is similar to conservation laws shown for fully connected networks. The conservation law shows a simple connection between the norms of weights of two consecutive layers. Using this law, an intuitive explanation is given for the trainability issues of deep GAT architectures. To mitigate this issue, a balanced initialization is suggested which shows improvements in generalization and training speed of deep GATs.


**Strengths:**

1. Novel conservation law for GAT architectures.
2. The paper and technical details are well written.


**Weaknesses:**

There are no major weaknesses.
1. The motivation for using the balanced initialization and why it should improve only for deep networks is not so clear.
2. The experimental results are not complete.

See Questions for more details.


**Questions:**

1. Does LLOrtho without balancing work well for deep networks? It is not clear if the cause for improvement is only balancing or also the initialization with orthogonal vectors.
2. The trainability issue is explained intuitively in pages 4-5:
(a) What is the reason that the trainability issue is amplified with depth? This is not explained.
(b) Why does this trainability issue occur only for GATs and not for fully connected networks? It seems that the same explanation holds for FC nets as well.
(c) Suggestion: It would be helpful to explain why the balanced initialization solves the trainability issue (it is not explained).


**Limitations:**

Limitations are not discussed. Maybe it would be helpful to add a section on this.

---

> ### Author Rebuttal · Authors · 2023-08-10
>
>
> We thank reviewer cXS5 for their insightful review and constructive questions, which are answered below. We would be happy to include the additional experiments and clarifications in the main paper as well.
>
> **1. Effect of orthogonal initialization:** We include results to compare the LLortho and LLortho+Bal initializations (See **Figure 1(a)** in additional PDF). We observe that balancing the LLortho initialization results in an improvement in the generalization ability of the network in most cases and speeds up training, particularly for deeper networks. However, note that the orthogonal initialization itself also has a positive effect on trainability in particular of deep networks. This follows from the comparison of Xavier+Bal and LLOrtho+Bal (or LLOrtho). Contributing to this fact is the mostly-balanced nature of the LLortho initialization i.e. given hidden layers of equal dimensions the network is balanced in all layers except the first and last layers, which allows the model to train, as opposed to the more severely imbalanced standard Xavier initialization. This further enforces the key takeaway of our work that norm imbalance at initialization hampers the trainability of GATs. In addition, the LLortho+Bal initialization also speeds up training over the LLortho initialization even in cases in which the generalization performance of the model is at par for both initializations.
>
> **2a): Amplification by depth:**
> The reason the trainability issue is amplified with depth can also be explained by our main theorem. Recursive substitution of Theorem 2.2 on the first term of the left side of Equation 8 (Page 4) results in a telescoping series yielding:
> $$ \sum_{j=1}^{n_{1}} \sum_{m=1}^{n_{0}} {W_{jm}^{(1)}}^2 \frac{\nabla_{W_{jm}^{(1)}} \mathcal{L}}{W_{jm}^{(1)}}  - \sum_{l=1}^{L-1} \sum_{o=1}^{n_{l}} {a_{o}^{(l)}}^2 \frac{\nabla_{a_{o}^{(l)}} \mathcal{L}}{a_{o}^{(l)}} =
> \sum_{i=1}^{n_{L-1}} \sum_{k=1}^{n_L} {W_{ki}^{(L)}}^2 \frac{\nabla{W_{ki}^{(L)}} \mathcal{L} }{W_{ki}^{(L)}} $$
> Note that $n_0$ is the input feature dimension. Generally, $2n_1 < n_0$ and thus $\mathbb{E} \left\lVert W^1[j:] \right\rVert ^2 = 2n_1/(n_1 + n_0) < 1$. Since the weights in the first layer and the gradients propagated to the first layer are both small, gradients of attention parameters of the intermediate hidden layers must also be very small in order to balance the equation. As the scalar product of attention parameters and gradients summed over the first and all intermediate layers must be less than the scalar product of weights and gradients of the first layer, the attention parameters and gradients in each intermediate layer do not have much room to change which hampers trainability. Evidently, the problem aggravates with depth where the same value must now be distributed over the parameters and gradients of more layers.
>
> **2b) Connection to fully-connected networks:**
> While it is true that a similar explanation as given in the paper holds also for fully connected networks (FCs), where the attention parameters are missing in Equation 8, the problem is more severe for GATs for two reasons:  i) Firstly, as explained above in (a), the summation of all intermediate layers of attention parameters is not a concern in FCs.
> ii) Secondly, FCs can achieve perfect dynamical isometry by employing an orthogonal looks-linear structure of feature weights, which enables signals to pass through very deep architectures [2]. However, due to the peculiarity of neighborhood aggregation in GATs (or GNNs), the same technique does not induce perfect dynamical isometry. Exploring how dynamical isometry can be achieved or approximated in general GNNs is an interesting follow-up direction of this work.
>
> **2c) Effect of balanced initialization:**
> On pages 4 and 5, we use Equation 8 to explain how norm imbalance is a cause of the trainability issue. We revisit this to analogously see how the balanced initialization mitigates the problem. A balanced initialization implies the weight norm of the last and second to the last layer are equal on both sides of the equation (as the attention parameters $a$ are set to 0). This allows larger relative gradients of the second to last layer (left side of the equation) (as compared to when the weights on the left were much larger than the weights on the right) which can enhance gradient flow in the network to earlier layers. In other words, gradients on both sides of the equation have equal room to drive parameter change.
>
> **3) Limitations:**
> We mention inline in the paper (line 159) that our theory defines a coarse-level conservation law, and thus cannot completely explain fine-grained training dynamics such as why the attention parameters change most in the first layer.  Secondly, the conservation law applies only to the self-attention defined in the original GAT and GATv2 models, and their architectural variations such as $\omega$GAT [3]. Note that the conservation law also holds for the non-attentive GCNs which are a special case of GATs (where the attention parameters $a$ are simply zero). Modeling different kinds of self-attention such as the dot-product self-attention in [4] entails modification of the conservation law, which has been left for future work. Following the reviewer's suggestion, we will dedicate an independent section to discuss these limitations.
>
> [1] Brody et al. How attentive are graph attention networks? In International Conference on Learning Representations, 2022.
>
> [2] Burkholz et al. Initialization of ReLUs for dynamical isometry. In Advances in Neural Information Processing Systems, volume 32, 2019.
>
> [3] Eliasof, et al. Improving Graph Neural Networks with Learnable Propagation Operators. In International Conference on Machine Learning, 2023.
>
> [4] Kim et al. How to find your friendly neighborhood: Graph attention design with self-supervision. In International Conference on Learning Representations, 2021.

---

### Author Rebuttal · Authors · 2023-08-10

We thank the reviewers for acknowledging the importance and relevance of our work and appreciate their constructive feedback, insightful questions, and encouraging comments. We take several actions to improve the paper in line with the reviewers' suggestions, summarized as follows:

i) We provide a more detailed explanation of some key ideas in the paper such as why the increasing depth of the network amplifies the trainability issue, how balancing the network mitigates the problem, and the effects of an orthogonal initialization.

ii) We conduct additional experiments as requested by the reviewers. Firstly, we include comparisons between imbalanced and balanced versions of Looks-Linear-Orthogonal, Identity, and Looks-Linear-Identity initializations. We also show the benefits of a balanced initialization on two other GNN models: the standard GCN and an architectural variation of GAT, the $\omega$GAT. We present these results in an additionally attached PDF and inline within individual responses to reviewers.

iii) We discuss the limitations of our work in more detail such as the need for modifications of the derived conservation law for other attention mechanisms such as self-attention in SuperGAT and Transformers.

We are grateful to the reviewers for their suggestions that enhance the presentation of our work. We will update our manuscript accordingly and would be happy to engage in discussions with the reviewers and answer any further questions.

---

### Decision · Program_Chairs · 2023-09-21

**Decision:**

Accept (poster)

**Comment:**

The reviewers appreciate the novel and solid technical contributions, and unanimously recommend acceptance and weak acceptance. The questions from the initial reviews were satisfactorily addressed.